# BRIDGING THE GAP TO REAL-WORLD OBJECT-CENTRIC LEARNING

**Maximilian Seitzer**[1,†]     **Max Horn**[2]     **Andrii Zadaianchuk**[1,3,†]     **Dominik Zietlow**[2]
**Tianjun Xiao**[2]     **Carl-Johann Simon-Gabriel**[2]     **Tong He**[2]     **Zheng Zhang**[2]
**Bernhard Schölkopf**[2]          **Thomas Brox**[2]          **Francesco Locatello**[2]
[1]Max-Planck Institute for Intelligent Systems, Tübingen, Germany
[2]Amazon Web Services
[3]Department of Computer Science, ETH Zürich

## ABSTRACT

Humans naturally decompose their environment into entities at the appropriate level of abstraction to act in the world. Allowing machine learning algorithms to derive this decomposition in an unsupervised way has become an important line of research. However, current methods are restricted to simulated data or require additional information in the form of motion or depth in order to successfully discover objects. In this work, we overcome this limitation by showing that reconstructing features from models trained in a self-supervised manner is a sufficient training signal for object-centric representations to arise in a fully unsupervised way. Our approach, `DINOSAUR`, significantly out-performs existing image-based object-centric learning models on simulated data and is the first unsupervised object-centric model that scales to real-world datasets such as COCO and PASCAL VOC. `DINOSAUR` is conceptually simple and shows competitive performance compared to more involved pipelines from the computer vision literature.

## 1 INTRODUCTION

Object-centric representation learning has the potential to greatly improve generalization of computer vision models, as it aligns with causal mechanisms that govern our physical world (Schölkopf et al., 2021; Dittadi et al., 2022). Due to the compositional nature of scenes (Greff et al., 2020), object-centric representations can be more robust towards out-of-distribution data (Dittadi et al., 2022) and support more complex tasks like reasoning (Assouel et al., 2022; Yang et al., 2020) and control (Zadaianchuk et al., 2020; Mambelli et al., 2022; Biza et al., 2022). They are in line with studies on the characterization of human perception and reasoning (Kahneman et al., 1992; Spelke & Kinzler, 2007). Inspired by the seemingly unlimited availability of unlabeled image data, this work focuses on *unsupervised* object-centric representation learning.

Most unsupervised object-centric learning approaches rely on a reconstruction objective, which struggles with the variation in real-world data. Existing approaches typically implement "slot"-structured bottlenecks which transform the input into a set of object representations and a corresponding decoding scheme which reconstructs the input data. The emergence of object representations is primed by the set bottleneck of models like Slot Attention (Locatello et al., 2020) that groups together independently repeating visual patterns across a fixed data set. While this approach was successful on simple synthetic datasets, where low-level features like color help to indicate the assignment of pixels to objects, those methods have failed to scale to complex synthetic or real-world data (Eslami et al., 2016; Greff et al., 2019; Burgess et al., 2019; Locatello et al., 2020; Engelcke et al., 2021).

To overcome these limitations, previous work has used additional information sources, e.g. motion or depth (Kipf et al., 2022; Elsayed et al., 2022). Like color, motion and depth act as grouping signals when objects move or stand-out in 3D-space. Unfortunately, this precludes training on most real-world

---

†: Work done during an internship at Amazon Web Services.
Correspondence to: `hornmax@amazon.de`, `maximilian.seitzer@tuebingen.mpg.de`

image datasets, which do not include depth annotations or motion cues. Following deep learning's mantra of scale, another appealing approach could be to increase the capacity of the Slot Attention architecture. However, our experiments (Sec. 4.3) suggest that scale alone is *not* sufficient to close the gap between synthetic and real-world datasets. We thus conjecture that the image reconstruction objective on its own does not provide sufficient inductive bias to give rise to object groupings when objects have complex appearance. But instead of relying on auxiliary external signals, we introduce an additional inductive bias by reconstructing features that have a high level of homogeneity within objects. Such features can easily be obtained via recent self-supervised learning techniques like DINO (Caron et al., 2021). We show that combining such a feature reconstruction loss with existing grouping modules such as Slot Attention leads to models that significantly out-perform other image-based object-centric methods and *bridge the gap to real-world object-centric representation learning*. The proposed architecture DINOSAUR (**DINO** and **S**lot **A**ttention **U**sing **R**eal-world data) is conceptually simple and highly competitive with existing unsupervised segmentation and object discovery methods in computer vision.

## 2 RELATED WORK

Our research follows a body of work studying the emergence of *object-centric representations* in neural networks trained end-to-end with certain architectural biases (Eslami et al., 2016; Burgess et al., 2019; Greff et al., 2019; Lin et al., 2020; Engelcke et al., 2020; Locatello et al., 2020; Singh et al., 2022a). These approaches implicitly define objects as repeating patterns across a closed-world dataset that can be discovered e.g. via semantic discrete- or set-valued bottlenecks. As the grouping of low-level features into object entities is often somewhat arbitrary (it depends for example on the scale and level of detail considered), recent work has explored additional information sources such as video (Kosiorek et al., 2018; Jiang et al., 2020; Weis et al., 2021; Singh et al., 2022b; Traub et al., 2023), optical flow (Kipf et al., 2022; Elsayed et al., 2022; Bao et al., 2022), text descriptions of the scene (Xu et al., 2022) or some form of object-location information (e.g. with bounding boxes) (Kipf et al., 2022). In contrast, we completely avoid additional supervision by leveraging the implicit inductive bias contained in the self-supervised features we reconstruct, which present a high level of homogeneity within objects (Caron et al., 2021). This circumvents the scalability challenges of previous works that rely on pixel similarity as opposed to perceptual similarity (Dosovitskiy & Brox, 2016) and enables object discovery on real-world data without changing the existing grouping modules. Our approach can be considered similar to SLATE (Singh et al., 2022a), but with the crucial difference of reconstructing *global* features from a Vision Transformer (Dosovitskiy et al., 2021) instead of *local* features from a VQ-VAE (van den Oord et al., 2017).

Challenging object-centric methods by *scaling dataset complexity* has been of recent interest: Karazija et al. (2021) propose ClevrTex, a textured variant of the popular CLEVR dataset, and show that previous object-centric models perform mostly poorly on it. Greff et al. (2022) introduce the MOVi datasets with rendered videos of highly realistic objects with complex shape and appearance. Arguably the most advanced synthetic datasets to date, we find that current state-of-the-art models struggle with them in the unsupervised setting. Finally, Yang & Yang (2022) show that existing image-based object-centric methods catastrophically fail on real-world datasets such as COCO, likely because they can not cope with the diversity of shapes and appearances presented by natural data. In contrast, we demonstrate that our approach works well on both complex synthetic and real-world datasets.

In the computer vision literature, structuring natural scenes without any human annotations has also enjoyed popularity, with tasks such as *unsupervised semantic segmentation* and *object localization*. Those tasks are interesting for us because they constitute established real-world benchmarks related to unsupervised object discovery, and we show that our method is also competitive on them. We refer to App. A for a detailed discussion of prior research in these areas.

## 3 METHOD

Our approach essentially follows the usual autoencoder-like design of object-centric models and is summarized in Figure 1: a first module extracts features from the input data (the encoder), a second module groups them into a set of latent vectors called *slots*, and a final one (the decoder) tries to reconstruct some target signal from the latents. However, our method crucially differs from other

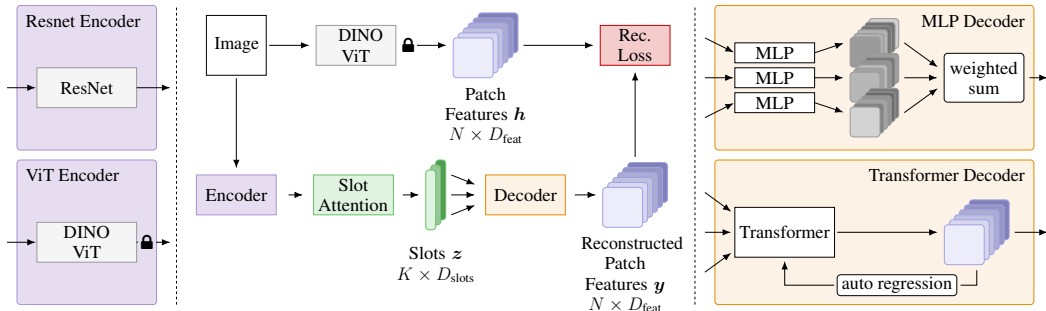

Figure 1: Overview of the proposed architecture `DINOSAUR`. The image is processed into a set of patch features $h$ by a frozen DINO ViT model (pre-trained using the self-supervised DINO method) and encoded via either a ResNet or the DINO ViT. Slot attention groups the encoded features into a set of slots. The model is trained by reconstructing the DINO features from the slots, either independently per-slot (MLP decoder) or jointly via auto regression (Transformer decoder).

approaches in that instead of reconstructing the original inputs, the decoder is tasked to reconstruct *features from self-supervised pre-training*. We start with the discussion of this training signal in Sec. 3.1 and describe further architectural choices in Sec. 3.2.

## 3.1 FEATURE RECONSTRUCTION AS A TRAINING SIGNAL

Why are models based on image reconstruction like Slot Attention not successful beyond simpler synthetic datasets? We hypothesize that reconstruction on the pixel level produces too weak of a signal for object-centricness to emerge; the task focuses (at least initially) strongly on low-level image features such as color statistics. This quickly decreases the reconstruction error, but the resulting model does not discover objects beyond datasets where objects are mostly determined by distinct object colors. Instead, if we had an (unsupervised) signal that required higher-level semantic information to reconstruct, there would be pressure on the slots to efficiently encode this information as well. Luckily, such signals can nowadays be easily obtained with self-supervised learning algorithms, which have been successful in learning powerful representations for vision tasks such as classification and object detection purely from images (Chen et al., 2020b; Grill et al., 2020; He et al., 2022). Thus, given $K$ slots $z \in \mathbb{R}^{K \times D_{\text{slots}}}$, the model is trained to reconstruct self-supervised features $h \in \mathbb{R}^{N \times D_{\text{feat}}}$, by minimizing the following loss:

$$\mathcal{L}_{\text{rec}} = \|y - h\|^2, \qquad y = \text{Decoder}(z). \tag{1}$$

This loss can be viewed as a form of student-teacher knowledge distillation (Hinton et al., 2015), where the student has a particular form of bottleneck that condenses the high-dimensional, unstructured information contained in the teacher features into a lower-dimensional, structured form. We can also draw parallels between this loss and perceptual similarity losses for image generation (Dosovitskiy & Brox, 2016), that is, the optimization takes place in a space more semantic than pixel space.

For pre-training, we utilize the ImageNet dataset (Deng et al., 2009). From the student-teacher perspective, this means that the teacher additionally transports knowledge gained from a larger image collection to the (smaller) datasets at hand. It is well-known that using large datasets for pre-training can significantly improve performance, but to our knowledge, we are the first to exploit such transfer learning for object-centric learning. In general, studying the role additional *data* can play for object-centric learning is an interesting topic, but we leave that for future investigations.

*Which self-supervised algorithm should we use?* In our analysis (Sec. 4.3), we investigate several recent ones (DINO (Caron et al., 2021), MoCo-v3 (Chen et al., 2021), MSN (Assran et al., 2022), MAE (He et al., 2022)). Interestingly, we find that they all work reasonably well for the emergence of real-world object grouping. In the following, we mainly apply the DINO method (Caron et al., 2021), because of its good performance and accessibility in open source libraries (Wightman, 2019). We experiment with features from ResNets (He et al., 2015) and Vision Transformers (ViTs) (Dosovitskiy et al., 2021), and find that the latter yield significantly better results.

## 3.2 AN ARCHITECTURE FOR REAL-WORLD OBJECT-CENTRIC LEARNING

**Encoder** Previous work has shown that powerful feature extractors help in scaling object-centric methods to more complex data (Kipf et al., 2022). To this end, we experiment with two choices: a ResNet-34 encoder with increased spatial resolution used by Kipf et al. (2022), and Vision Transformers. Unfortunately, we were not able to optimize randomly initialized ViTs with our model, as training collapsed. Instead, we found it sufficient to initialize the ViT using weights from self-supervised pre-training, and keeping them fixed throughout training[1]. In terms of results, we find that the ResNet and the pre-trained ViT encoder perform similarly. However, the model converges faster with the pre-trained ViT, and it is also computationally more efficient: we can directly use the ViT outputs as the target features $h$. Consequently, we mainly use the ViT encoder in the following.

**Slot Attention Grouping** The grouping stage of our model uses Slot Attention (Locatello et al., 2020) to turn the set of encoder features into a set of $K$ slot vectors $z \in \mathbb{R}^{K \times D_{\text{slots}}}$. This follows an iterative process where slots compete for input features using an attention mechanism, starting from randomly sampled initial slots. We largely use the original Slot Attention formulation (including GRU (Cho et al., 2014) and residual MLP modules), with one difference when using ViT features: we do not add positional encodings on the ViT features before Slot Attention, as we found the ViT's initial position encodings to be sufficient to support spatial grouping of the features. Additionally, we add a small one-hidden-layer MLP that transforms each encoder feature before Slot Attention.

**Feature Decoding** As we apply feature instead of image reconstruction as the training objective, we need a decoder architecture suitable for this purpose. To this end, we consider two different designs: a MLP decoder that is applied independently to each slot, and a Transformer decoder (Vaswani et al., 2017) that autoregressively reconstructs the set of features. We describe both options in turn.

The *MLP decoder* follows a similar design as the commonly used spatial broadcast decoder (Watters et al., 2019). Each slot is first broadcasted to the number of patches, resulting in a set of $N$ tokens for each slot. To make the spatial positions of the tokens identifiable, a learned positional encoding is added to each token. The tokens for each slot are then processed token-wise by the same MLP, producing the reconstruction $\hat{y}_k$ for slot $k$, plus an alpha map $\alpha_k$ that signifies where the slot is active. The final reconstruction $y \in \mathbb{R}^{N \times D_{\text{feat}}}$ is formed by taking a weighted sum across the slots:

$$y = \sum_{k=1}^{K} \hat{y}_k \odot m_k, \qquad m_k = \operatorname*{softmax}_k \alpha_k \qquad (2)$$

The advantage of this simple design is its computational efficiency: as the MLP is shared across slots and positions, decoding is heavily parallelizable.

The *Transformer decoder* (Vaswani et al., 2017) reconstructs features $y$ jointly for all slots in an autoregressive manner. In particular, the feature at position $n$ is generated while conditioning on the set of previously generated features $y_{<n}$ *and* the set of slots $z$: $y_n = \text{Decoder}(y_{<n}; z)$. This decoder design is more powerful than the MLP decoder as it can maintain global consistency across the reconstruction, which might be needed on more complex data. However, we found several drawbacks of the Transformer decoder: it does not work with training ResNet encoders from scratch, higher resolution target features (see App. D.5), and requires more effort to tune (see App. D.4). Thus, we recommend using the MLP decoder as the first choice when applying DINOSAUR to a new dataset. We note that Transformer decoders have also been previously explored by SLATE (Singh et al., 2022a) and STEVE (Singh et al., 2022b), but to reconstruct the discrete token map of a VQ-VAE (van den Oord et al., 2017).

**Evaluation** Object-centric methods are commonly evaluated by inspecting masks associated with each slots. Previous approaches reconstructing to image-level typically use the decoder's alpha mask for this purpose; for the MLP decoder, we also make use of this option. The Transformer decoder does not produce an alpha mask. Instead, we have two options: the attention masks of Slot Attention (used by SLATE), or the decoder's attention mask over the slots. We found that the latter performed better (see Sec. D.6), and we use it throughout. As the masks from feature reconstruction are of low resolution, we bilinearly resize them to image resolution before comparing them to ground truth masks.

---

[1]Another option would be further finetuning the pre-trained ViT, but we found that this leads to slots that do not focus on objects. Combining ViT training with Slot Attention might require very careful training recipes.

## 4 EXPERIMENTS

Broadly, we pursue two goals with our experiments: 1) demonstrating that our approach significantly extends the capabilities of object-centric models towards real-world applicability (Sec. 4.1), and 2) showing that our approach is competitive with more complex methods from the computer vision literature (Sec. 4.2). Additionally, we ablate key model components to find what is driving the success of our method (Sec. 4.3). The main task we consider in this work is *object discovery*, that is, finding pixel masks for all object instances in an image.

**Datasets**   We consider two synthetic and two real-world image datasets. As synthetic datasets, we use the MOVi datasets (Greff et al., 2022), recently introduced as challenging testbeds for object-centric methods. In particular, we use the variants MOVi-C and MOVi-E, which contain around 1 000 realistic 3D-scanned objects on HD backgrounds. For our purposes, the main difference is that MOVi-C contains 3–10, and MOVi-E 11–23 objects per scene. Note that we treat the video-based MOVi datasets as image datasets by randomly sampling frames. As real-world datasets, we use PASCAL VOC 2012 (Everingham et al., 2012) and MS COCO 2017 (Lin et al., 2014), commonly used for object detection and segmentation. Whereas PASCAL VOC contains many images with only a single large object, COCO consists of images with at least two and often dozens of objects. Both datasets represent a significant step-up in complexity to what object-centric models have been tested on so far. In App. B.2, we also report preliminary results on the KITTI driving dataset.

**Training Details**   We train `DINOSAUR` using the Adam optimizer (Kingma & Ba, 2015) with a learning rate of $4 \cdot 10^{-4}$, linear learning rate warm-up of 10 000 optimization steps and an exponentially decaying learning rate schedule. Further, we clip the gradient norm at 1 in order to stabilize training and train for 500k steps for the MOVI and COCO datasets and 250k steps for PASCAL VOC. The models were trained on 8 NVIDIA V100 GPUs with a local batch size of 8, with 16-bit mixed precision. For the experiments on synthetic data, we use a ViT with patch size 8 and the MLP decoder. For the experiments on real-world data, we use a ViT with patch size 16 and the Transformer decoder. We analyze the impact of different decoders in Sec. 4.3. The main results are averaged over 5 random seeds; other experiments use 3 seeds. Further implementation details can be found in App. E.1.

### 4.1 COMPARISON TO OBJECT-CENTRIC LEARNING METHODS

Our goal in this section is two-fold: 1) demonstrating that previous object-centric methods fail to produce meaningful results on real-world datasets and struggle even on synthetic datasets, and 2) showcase how our approach of incorporating strong pre-trained models results in a large step forward for object-centric models on both kinds of datasets.

**Tasks**   We evaluate on the task object-centric models are most frequently tested on: object discovery (Burgess et al., 2019), that is, producing a set of masks that cover the independent objects appearing on an image. We also present preliminary results testing the quality of the learned representations on the COCO dataset in App. B.3, though this is not the main focus of our work.

**Metrics**   As common in the object-centric literature, we evaluate this task using foreground adjusted rand index (FG-ARI), a metric measuring cluster similarity. Additionally, we compute a metric based on intersection-over-union (IoU), the mean best overlap (mBO) (Pont-Tuset et al., 2017). mBO is computed by assigning each ground truth mask the predicted mask with the largest overlap, and then averaging the IoUs of the assigned mask pairs. In contrast to ARI, mBO takes background pixels into account, thus also measuring how close masks fit to objects. On datasets where objects have a semantic label attached (e.g. on COCO), we can evaluate this metric with instance-level (i.e. object) masks, and semantic-level (i.e. class) masks. This allows us to find model preferences towards instance- or semantic-level groupings.

**Baselines**   We compare our approach to a more powerful version of Slot Attention (Locatello et al., 2020) based on a ResNet encoder that has been shown to scale to more complex data (Elsayed et al., 2022). Further, we compare with SLATE (Singh et al., 2022a), a recent object-centric model that trains a discrete VQ-VAE (van den Oord et al., 2017) as the feature extractor and a Transformer as the decoder. We refer to App. E.2 for details about baseline configurations.

As it can be hard to gauge how well object-centric methods perform on new datasets solely from metrics, we add one trivial baseline: dividing the image into a set of regular blocks. These *block masks*

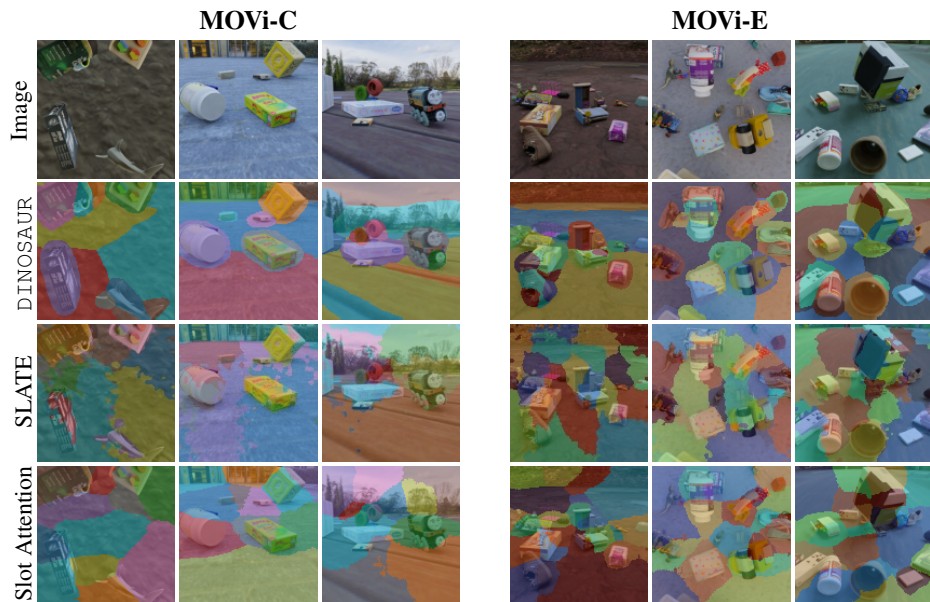

Figure 2: Example results on the synthetic MOVi-C and MOVi-E datasets (Greff et al., 2022).

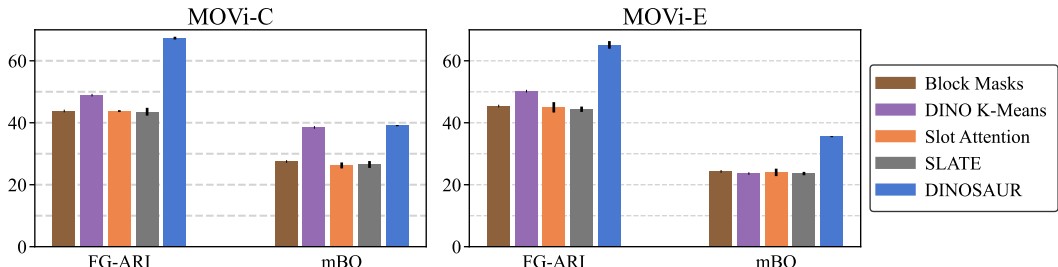

Figure 3: Object Discovery on synthetic datasets (mean $\pm$ standard dev., 5 seeds) with 11 (MOVi-C) and 24 slots (MOVi-E). We report foreground adjusted rand index (FG-ARI) and mean best overlap (mBO). DINOSAUR uses a ViT-B/8 encoder with the MLP decoder.

(see Fig. 18) thus show the performance of a method that only follows a geometric strategy to group the data, completely ignoring the semantic aspects of the image. Familiar to practitioners, this is a common failure mode of object-centric methods, particularly of Slot Attention. Last, we apply the *K-Means algorithm* on the DINO features and use the resulting clustering to generate spatial masks. This baseline shows to which extent objects are already trivially extractable from self-supervised features.

**Results on Synthetic Datasets (Fig. 2 and Fig. 3)**  Both Slot Attention and SLATE struggle on the challenging MOVi datasets, performing similar to the naive block masks and worse than the K-Means baselines. Our model achieves good performance on both MOVI-C and MOVi-E. In App. B.1, we also find that our method compares favorably to video methods that can use temporal information and/or weak supervision (Elsayed et al., 2022; Singh et al., 2022b)

**Results on Real-World Datasets (Fig. 4 and Fig. 5)**  As expected, Slot Attention can not handle the increased complexity of real-world data and degrades to non-semantic grouping patterns. For SLATE, semantic grouping begins to emerge (e.g. of backgrounds), but not consistently; it still performs worse than the K-Means baseline. Note that it was necessary to early-stop SLATE's training as performance would degrade to Slot Attention's level with more training. In contrast, DINOSAUR captures a variety of objects of different size, appearance and shape. To the best of our knowledge, we are the first to show a successful version of an object-centric model on unconstrained real-world images in the fully unsupervised setting. Our result represents a significant step-up in complexity of what object-centric

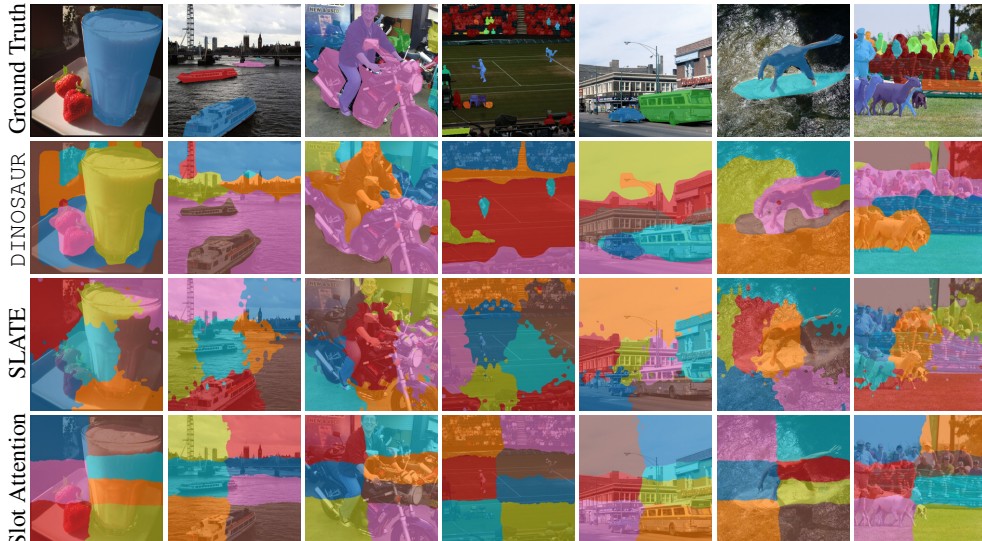

Figure 4: Example reults on COCO 2017, using 7 slots. Additional examples are provided in App. G.

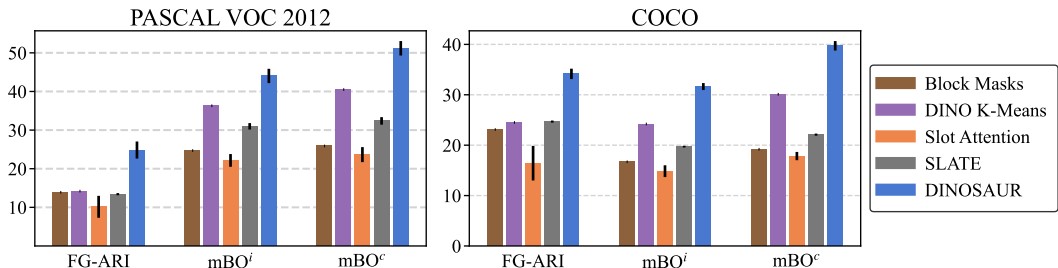

Figure 5: Object Discovery on real-world datasets (mean ± standard dev., 5 seeds) with 6 (PASCAL) and 7 slots (COCO). We report foreground adjusted rand index (FG-ARI) and instance/class mean best overlap (mBO$^i$/mBO$^c$). DINOSAUR uses a ViT-B/16 encoder with the Transformer decoder.

methods can handle. Note that the examples in Fig. 4 show mostly semantic rather than instance grouping emerging: this is a by-product of using the Transformer decoder. In contrast, the MLP decoder is biased towards instance grouping, an effect which we analyze in Sec. 4.3.

## 4.2 COMPARISON TO COMPUTER VISION METHODS

In this section, our goal is to show that our method fares well on two tasks closely related to object discovery from the computer vision literature: unsupervised object localization and segmentation. Being competitive on these benchmarks is difficult, as there has been a stream of methods with quickly improving results recently (Wang et al., 2022; Hamilton et al., 2022; Zadaianchuk et al., 2023). Due to space issues, we defer most of the discussion to App. C.

**Tasks, Metrics and Baselines** We briefly introduce the two tasks: in *object localization*[2], the goal is to find object location and size by predicting bounding boxes, and in *unsupervised semantic segmentation* the goal is to separate the image into semantically consistent labeled regions. For the latter, we consider two variations: object segmentation, where only foreground objects should get segmented and labeled, and scene decomposition, where each pixel of the image has to be labeled with a semantic class. We evaluate object localization in terms the fraction of images on which at least one object was correctly localized (CorLoc) (Vo et al., 2020), and semantic segmentation in terms of mean intersection-over-union over classes (mIoU). For semantic segmentation, we obtain class

---

[2]This task is often called "object discovery" in the literature as well, but we term it "object localization" in this work in order to avoid confusion with the task evaluated in the object-centric literature.

Table 1: Representative comparisons on three tasks from the computer vision literature. We refer to App. C for a detailed discussion including more datasets, baselines, and metrics. Here, we compare with (a) DeepSpectral (Melas-Kyriazi et al., 2022) and TokenCut (Wang et al., 2022), (b) MaskContrast (Van Gansbeke et al., 2021) and COMUS (Zadaianchuk et al., 2023), and (c) SlotCon (Wen et al., 2022) and STEGO (Hamilton et al., 2022). DINOSAUR uses a ViT-B/16 encoder with the Transformer decoder (mean $\pm$ standard dev., 5 seeds).

| (a) Unsup. Object Localization. | | (b) Unsup. Object Segmentation. | | (c) Unsup. Scene Decomposition. | |
|---|---|---|---|---|---|
| **COCO-20k (CorLoc)** | | **PASCAL VOC 2012 (mIoU)** | | **COCO-Stuff 27 (mIoU)** | |
| DeepSpectral | 52.2 | MaskContrast | 35.0 | SlotCon | 18.3 |
| TokenCut | 58.8 | COMUS | 50.0 | STEGO | 26.8 |
| DINOSAUR | 67.2 $\pm$1.5 | DINOSAUR | 37.2 $\pm$1.8 | DINOSAUR | 24.0 $\pm$0.9 |

labels by running K-Means clustering on features associated with each slot after training the model, then assigning clusters to ground truth classes by maximizing IoU using Hungarian matching, similar to Van Gansbeke et al. (2021) (see App. C for details). On each task, we compare with the current state-of-the-art (Wang et al., 2022; Hamilton et al., 2022; Zadaianchuk et al., 2023), and a recent, but competitive method (Van Gansbeke et al., 2021; Melas-Kyriazi et al., 2022; Wen et al., 2022).

**Results (Table 1)**    For object localization, our method reaches comparable results to what has been previously reported. For object segmentation, our method falls behind the state-of-the-art, though it is still competitive with other recent work. Note that the best methods on this task employ additional steps of training segmentation networks which improves results and allows them to run at the original image resolution. In contrast, the masks we evaluate are only of size $14 \times 14$; we leave it to future work to improve the resolution of the produced masks. For the task of scene decomposition, DINOSAUR comes close to the current state-of-the-art. All in all, our method is competitive with often more involved methods on these benchmarks, demonstrating a further step towards real-world usefulness of object-centric methods.

## 4.3 ANALYSIS

In this section, we analyze different aspects of our approach: the importance of feature reconstruction, the impact of the method for self-supervised pre-training, and the role of the decoder. Additional experiments are included in App. D.

**Insufficiency of Image Reconstruction**    We first test the hypothesis if a scaled-up Slot Attention model trained with image reconstruction could lead to real-world object grouping. Our experiments from Sec. 4.1 already show that a ResNet encoder is not sufficient. We additionally test a ViT-B/16 encoder under different training modes: training from scratch, frozen, or finetuning DINO pre-trained weights. We find that training from scratch results in divergence of the training process, and that both the frozen and finetuning setting fail to yield meaningful objects, resulting in striped mask patterns (see Fig. 12). Thus, even when starting from features that are highly semantic, image reconstruction does not give enough signal towards semantic grouping.

**ResNet and Pre-Trained ViT Encoders Perform Similar**    Second, we analyze whether *pre-training the encoder* plays a crucial role for our method. To do so, we compare ResNet34 encoders trained from scratch with pre-trained ViT encoders, and find that the performance is overall similar (see Table 12). This also suggests that the feature reconstruction signal is the key component in our approach that allows object-centricness to emerge on real-world data. We expand in App. D.2.

**Choice of Self-Supervised Targets (Table 2)**    We now analyze the role of the self-supervised pre-training algorithm. To this end, we train DINOSAUR with a ResNet34 encoder (from scratch) on COCO, but reconstruct targets obtained from ViTs pre-trained with different methods: DINO, MoCo-v3, MSN, and MAE. Remarkably, all self-supervised schemes perform well for the task of object discovery (examples in Fig. 24). This demonstrates that self-supervised pre-training on ImageNet translates into a useful, general bias for discovering objects.

**Choice of Decoder (Table 3)**    We compare the choice of MLP vs. Transformer decoder for object discovery. Both options use a ViT-B/16 encoder. Generally, we find that the MLP decoder is better on ARI whereas the Transformer decoder is better on mBO. For MOVi-C, visual inspection (see Fig. 19)

Table 2: Comparing self-supervised reconstruction targets produced by a ViT-B/16 on COCO object discovery, with a ResNet34 encoder and the MLP decoder.

| Algorithm | FG-ARI | mBO$^i$ | mBO$^c$ |
|---|---|---|---|
| DINO | 40.9 ±0.2 | 27.9 ±0.0 | 31.1 ±0.1 |
| MoCo-v3 | 40.4 ±0.6 | 28.1 ±0.2 | 31.1 ±0.1 |
| MSN | 40.7 ±0.3 | 27.6 ±0.1 | 30.7 ±0.1 |
| MAE | 37.7 ±0.1 | 28.1 ±0.1 | 31.7 ±0.0 |

Table 3: Comparing different decoders on object discovery, with a ViT-B/16 encoder. We also list mean squared reconstruction error (MSE).

| Dataset | Decoder | ARI | mBO$^{(i,c)}$ | | MSE |
|---|---|---|---|---|---|
| MOVi-C | MLP | 66.0 | 35.0 | | 0.24 |
| | Transformer | 55.7 | 42.4 | | 0.14 |
| PASCAL | MLP | 24.6 | 39.5 | 40.9 | 0.33 |
| | Transformer | 24.8 | 44.0 | 51.2 | 0.17 |
| COCO | MLP | 40.5 | 27.7 | 30.9 | 0.31 |
| | Transformer | 34.1 | 31.6 | 39.7 | 0.16 |

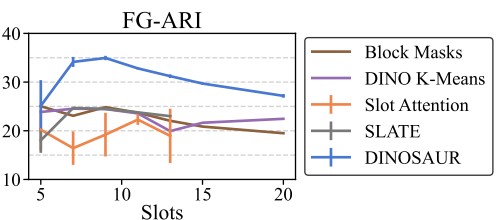

Figure 6: Sensitivity to number of slots on COCO object discovery (see also App. D.3).

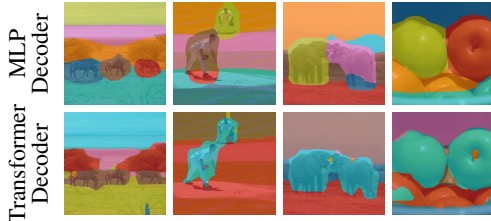

Figure 7: MLP and Transformer decoder have different biases in how they group objects.

shows that the Transformer tends to produce tighter masks and a cleaner background separation, but uses excess slots to split objects. For PASCAL and COCO, what's striking is the Transformer decoders' improvement of 9–10 class mBO. This reveals that the Transformer decoder is biased towards grouping semantically related instances into the same slot, which we suggest stems from its global view on the image, but also from the generally increased expressiveness of the architecture (cf. the lower reconstruction loss). In contrast, the MLP decoder is able to separate instances better (see Fig. 7 and also Fig. 23), which is reflected in the higher ARI scores. Researching how different decoder designs affect semantic vs. instance-level grouping is an interesting avenue for future work. In App. D.4 and App. D.5, we further study different decoder properties.

## 5 CONCLUSION

We presented the first image-based fully unsupervised approach for object-centric learning that scales to real-world data. Our experiments demonstrate significant improvements on both simulated and real-world data compared to previously suggested approaches and even achieve competitive performance with more involved pipeline methods from the computer vision literature.

This work only takes a first step towards the goal of representing the world in terms of objects. As such, some problems remain open. One issue concerns semantic vs. instance-level grouping. As evident from the presented examples, our approach covers a mix of both, with semantically related objects sometimes being grouped into a single slot. While we found the type of decoder to influence this behavior, more fine-grained control is needed. A related issue is the detail of the decomposition, e.g. whether objects are split into parts or stay whole. We found this to be dependent on the number of slots, with a fixed number often being inappropriate (see Fig. 6 and App. D.3). How models can dynamically choose a suitable level of detail while staying unsupervised but controllable will be an important challenge to fully master the ambiguities the real world inherently presents.

In this work, we mainly focused on object discovery. Future work could further examine the properties of the learned slot representations, for instance robustness to distribution shifts, generalization and usefulness for downstream tasks. Another interesting direction is how our approach can be combined with image generation to build flexible and compositional generative models of natural data.

## REPRODUCIBILITY STATEMENT

Appendix E.1 contains detailed information about the `DINOSAUR` architecture and all hyperparamers used for all experiments. Appendix E.2 contains details about how baselines were trained. Appendix F contains information about task evaluation and datasets. All datasets used in this work (MOVi, PASCAL VOC 2012, COCO, KITTI) are public and can be obtained on their respective web pages. Source code will be made available under `https://github.com/amazon-science/object-centric-learning-framework`.

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

# Appendix

## Table of Contents

## A    EXTENDED RELATED WORK

**Unsupervised Semantic Segmentation** is the challenging task of structuring natural scene into semantically coherent regions without any human annotations. Methods for this task (Ji et al., 2019; Ouali et al., 2020; Cho et al., 2021; Van Gansbeke et al., 2021; 2022; Hamilton et al., 2022; Zadaianchuk et al., 2023) are typically separated into two independent steps: first, representations for each pixel by a pre-trained self-supervised model are refined; second, these representations are clustered on the whole dataset to assign a class category to each pixel. DINOSAUR combines those steps into a simple method that groups semantically meaningful regions jointly with learning region representations.

Approaches that try to learn and cluster dense representations without additional assumptions (Ji et al., 2019; Ouali et al., 2020) were shown to be effective only on small-scale datasets with narrow visual domains. More advanced methods incorporate geometric consistency (Cho et al., 2021) or unsupervised saliency detection (Van Gansbeke et al., 2022; Zadaianchuk et al., 2023) to simplify dense representation learning and the grouping task. Additionally, using self-supervised DINO representations (Caron et al., 2021) by contrasting them (Hamilton et al., 2022) or to self-train a semantic segmentation model (Van Gansbeke et al., 2022; Melas-Kyriazi et al., 2022; Zadaianchuk et al., 2023) allows obtaining a more accurate image decomposition to semantic categories.

**Object Localization** is another research direction that aims at structuring the scene from images, but by predicting object-level features such as bounding boxes or category. Recent works (Vo et al., 2020; 2021; Simeoni et al., 2021; Melas-Kyriazi et al., 2022; Wang et al., 2022) often group pre-trained, self-supervised features of individual images into initial object proposals, and then aggregate and cluster these proposals across images to assign them labels. Self-training can then be used on top to achieve multi-object detection on natural scenes.

At first, rOSD (Vo et al., 2020; 2021) used features obtained from a supervised classifier to localize salient objects in the image. Next, LOST (Simeoni et al., 2021) showed that dense, self-supervised DINO features can also be used for localization of the most prominent object in the image. Finally, DeepSpectral (Melas-Kyriazi et al., 2022) and TokenCut (Wang et al., 2022) showed that additionally transforming DINO features to a spectral embedding can significantly improve the quality of the object proposals. While those methods can be combined with self-training of object detectors like R-CNN, they are limited by the quality of the set of salient objects used as pseudo-labels. In contrast, DINOSAUR splits each scene to a set of object proposals enabling multi-object scene decomposition without an additional recognition network.

**Architectures making use of slot-like components** have also enjoyed popularity aside from the research explicitly focused on object-centric learning. For instance, the Perceiver model learns to distill a large set of input tokens from different modalities into a reduced set of embeddings, which can be seen as slots (Jaegle et al., 2022). Prominent recent successes of large-scale multi-modal modeling such as Flamingo (Alayrac et al., 2022) or Gato (Reed et al., 2022) are based on the Perceiver architecture. In dynamics modeling, recurrent independent mechanisms (Goyal et al., 2021) use a competition between independent modules to read inputs into a slot-structured latent state.

Recent work on contrastive learning such as SlotCon (Wen et al., 2022) or Odin (Hénaff et al., 2022) also can be seen as learning slot representations, although with a focus on semantic (class-based) slots rather than instances. In the case of SlotCon, this is due to the reliance on prototypes which naturally lead to semantically focused representations whereas in the case of Odin, the contrastive objective does not encourage differentiation of one or multiple instances. Further, Odin is positioned as a strategy to pre-train a backbone which can then be finetuned on downstream tasks and is not intended to be an unsupervised method that directly leads to instance-level grouping. Note that even though Odin also evaluates object discovery on COCO using the mBO metric, the obtained numbers are not comparable to ours as Odin generates 255 proposal masks by running K-Means 128 times with different numbers of clusters. Finally, ORL (Xie et al., 2021) addresses the reliance of many contrastive methods on ImageNet-specific biases, which leads to intra-image contrasting of the same object under different augmentations. This is done by identifying correspondences of objects in different images using models trained via intra-image contrasting and subsequently training the model using inter-image contrasting by selecting objects of the same type from different images.

Table 4: Object discovery on synthetic datasets (mean ± standard dev., 5 seeds), corresponding to results from Fig. 3. We additionally add results (over 3 seeds) for the video methods STEVE (Singh et al., 2022b) and SAVi++ (Elsayed et al., 2022). Note that this version of SAVi++ is unconditional, i.e. it does not have access to the objects of the first frame. We train SAVi++ with (unsupervised) image reconstruction, and (supervised) optical flow reconstruction. To ensure a valid comparison, STEVE and SAVi++ are trained on video, but evaluated by processing each video frame independently.

| | MOVi-C | | MOVi-E | |
|---|---|---|---|---|
| | FG-ARI | mBO | FG-ARI | mBO |
| *Block Masks* | *43.8* | *27.5* | *45.4* | *24.3* |
| *DINO K-Means* | *48.9* | *38.5* | *50.2* | *23.6* |
| Slot Attention | 43.8 ±0.3 | 26.2 ±1.0 | 45.0 ±1.7 | 24.0 ±1.2 |
| SLATE | 43.6 ±1.3 | 26.5 ±1.1 | 44.4 ±0.9 | 23.6 ±0.5 |
| DINOSAUR (ViT-S/8) | 67.2 ±0.3 | 38.6 ±0.1 | 64.7 ±0.7 | 34.1 ±0.1 |
| DINOSAUR (ViT-B/8) | 68.6 ±0.4 | 39.1 ±0.2 | 65.1 ±1.2 | 35.5 ±0.2 |
| STEVE | 44.3 ±0.8 | | 44.5 ±2.8 | |
| SAVi++ (Image Recon.) | 10.4 ±0.1 | | 15.9 ±1.1 | |
| SAVi++ (Flow Recon.) | 30.0 ±1.5 | | 19.8 ±2.0 | |

# B  EXPANDED COMPARISON WITH OBJECT-CENTRIC METHODS

In this section, we present additional results on object discovery. In particular, we compare to two state-of-the-art video-based object-centric methods: STEVE (Singh et al., 2022b) and SAVi++ (Elsayed et al., 2022) (Sec. B.1). We also present preliminary results on the KITTI driving dataset (Alhaija et al., 2018) (Sec. B.2), and a study of the learned slot representations on COCO (Sec. B.3).

## B.1  COMPARISON TO VIDEO METHODS

Having access to motion information is often thought to be advantageous for discovering objects (Greff et al., 2020; Kipf et al., 2022), as object identity and motions are stable through time. It is thus interesting how our method compares against models that have been trained on videos. To this end, we test two recent video-based object-centric: STEVE (Singh et al., 2022b) and SAVi++ (Elsayed et al., 2022). STEVE is an extension of SLATE (Singh et al., 2022a) to videos, whereas SAVi++ is a scaled-up version of SAVi (Kipf et al., 2022), which in turn is the video extension of Slot Attention (Locatello et al., 2020). We train both of them on the synthetic MOVi-C (with 11 slots) and MOVi-E (with 24 slots), using the original hyperparameters proposed for those datasets. However, we train SAVi++ unconditionally, that is, using random learnt slot initializations instead of providing first-frame supervision of the object positions. We also had to reduce the batch size of SAVi++ from 64 to 32 due to computational reasons. STEVE is trained for 200k steps, and SAVi++ for 300k steps. To ensure a valid comparison to the image-based methods, we evaluate FG-ARI on the individual frames of the MOVi datasets, reinitializing slots for each frame.

**Results (Table 4)**    Somewhat surprisingly, the video-based methods do not perform better than the image-based methods. For both STEVE and SAVi++, there is a large gap to our method. STEVE performs similar to its image-based counterpart SLATE. Compared to the results reported by Elsayed et al. (2022), SAVi++ performs considerably worse, falling behind its image counterpart Slot Attention. Even when having access to optical flow supervision, SAVi++ does not produce meaningful object groupings. We hypothesize this is because SAVi++ was not designed for the unconditional setting, and model and/or hyperparameter changes are needed to alleviate this.

## B.2  RESULTS ON THE KITTI DRIVING DATASET

We present preliminary results on KITTI (Alhaija et al., 2018), consisting of real-world videos of city-street driving. In particular, we evaluate on the instance segmentation annotations, containing 200 independently annotated frames. For training, we use all 147 videos by randomly sampling frames from them, resulting in 95 869 training images. We compare with "Discovering Objects that Can Move" (Bao et al., 2022), a recent video-based object-centric method that uses motion masks to

supervise the attention masks. To be consistent, we evaluate at the same mask resolution as them, namely at $1242 \times 375$ pixels.

Because of the skewed aspect ratio and high resolution, and because our model is trained on square images, we evaluate in a sliding window fashion. In particular, we partition the image into 4 non-overlapping windows of size $375 \times 375$ (adding padding of 129 black pixels to both sides), process each image individually into $K$ masks of size $375 \times 375$ using our model, then join the masks spatially and removing padding. See Fig. 8 for an example window. We use the resulting $4 \cdot K$ masks of size $1242 \times 375$ for evaluation. This kind of sliding window approach has the obvious flaw that objects can not span more than window. We leave it to future work to improve working with high-resolution images.

Tuning DINOSAUR to work on KITTI was more difficult than for COCO or PASCAL VOC. For instance, using the Transformer decoder did not result in object groupings, and we had to resort to the MLP decoder with a small slot size of 32. We also had difficulties training a ResNet encoder on KITTI. We hypothesize this is because the KITTI dataset does not have a lot of diversity, and so high-capacity architectures can easily overfit to the expected scene layout instead of learning to discover objects.

**Results (Table 5)** We train the model with $K = 9$ slots, yielding 36 slots for evaluation. Our model clearly outperforms Bao et al. (2022)'s method in terms of FG-ARI, even though it is trained with motion supervision. However, there is ample room for improvement: visual inspection shows that while our method manages to capture larger objects in the foreground (mainly cars), the many small objects in the background of the scene are grouped together with non-objects, e.g. houses. This explains why FG-ARI is relatively high (as large objects influence this metric more), while mBO is relatively low (as the metric is averaged over *all* objects). This hints at another difficulty of real-world data which current object-centric methods are not well equipped to deal with: diversity of object scales.

Table 5: Object discovery on KITTI (mean $\pm$ standard dev., 5 seeds).

| Method | FG-ARI | mBO |
|---|---|---|
| Bao et al. (2022) | 47.1 | |
| DINOSAUR | 70.3 $\pm$0.8 | 19.4 $\pm$0.3 |

Figure 8: Example masks on KITTI.

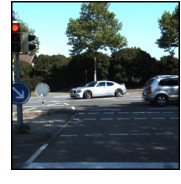 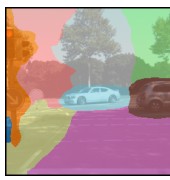

### B.3 OBJECT PROPERTY PREDICTION ON COCO

We study the usefulness of the learned representations for downstream applications by predicting *object properties* from the slots on COCO. To this end, we supervise shallow predictors using the slot representations from unsupervised object-centric models. As properties, we use the class, and the x- and y-coordinates of each object (center-of-mass).

We closely follow the experimental design of Dittadi et al. (2022), and did not attempt to tune their settings to optimize for performance. As downstream models, we test a linear predictor, and a one-hidden layer MLP. Each model is applied in parallel to all slots for an image, i.e. it receives a slot as input, and outputs a vector of property predictions for this slot. We use the cross-entropy loss for the categorical class prediction, and the MSE loss for the continuous x- and y-coordinates, where the overall loss is just the sum of the individual losses. We match the predicted set of vectors to the target set of vectors using the Hungarian method (Kuhn, 1955), minimizing the total loss of the assignment. As metrics, we report Top-1 and Top-5 average-class accuracy for the 80-way class prediction and $R^2$ (coefficient of determination) for the x- and y-coordinates.

We use 10 000 random images from the COCO dataset to train, and another 1 000 random images as a validation set. We report metrics on the 5 000 validation images of COCO. Images that contain no objects due to center cropping are filtered out. As COCO images can contain many objects, but our method typically only uses a low number of slots (e.g. 7), we keep only the 6 largest objects in the image. This way, all ground truth objects can in principle be matched by a slot. We train for 15 000 steps, and use the model with the lowest validation loss for testing.

The results are listed in Fig. 9. We compare DINOSAUR using a DINO-pretrained ViT-B/16 encoder with a Slot Attention model, all using 7 slots. On class prediction, our method has 25% higher accuracy than Slot Attention. On coordinate prediction, DINOSAUR reaches similar performance to Slot Attention on the x-coordinate, but exceeds it on the y-coordinate, where Slot Attention drops-off.

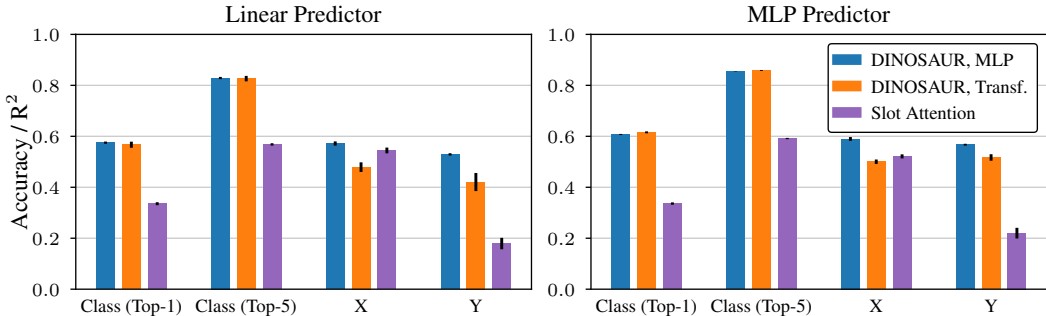

Figure 9: *Object property prediction* on the validation set of COCO, showing mean and standard deviation over 3 seeds. We train a linear predictor (left) and a one-hidden layer MLP (right) to predict object properties from the slot representations. For class prediction, we show Top-1 and Top-5 accuracy. For coordinate prediction, we show $R^2$ (coefficient of determination, higher is better). We compare our method using MLP and Transformer decoders with Slot Attention.

We also find that on coordinate prediction, the representations from the Transformer decoder are a bit worse than from the MLP decoder. This may be because the Transformer decoder often captures several disconnected objects in one slot, making it more difficult to predict the center-of-mass. Using a non-linear instead of a linear predictor slightly improves accuracy on all properties; using more hidden layers than one did not improve the performance further in our experiments. Overall, we conclude that the representation DINOSAUR learns are more useful than Slot Attention's for downstream tasks on real-world data. However, our method also still leaves room for improvement. Finally, we note that this experiment only constitutes a first step in evaluating the learned representations, and that further investigations could be conducted, e.g. taking into account robustness to distribution shifts.

## C  EXPANDED COMPARISON WITH COMPUTER VISION METHODS

In this section, we extend the comparison to computer vision methods with a more detailed discussion. In addition to the results from Sec. 4.2, we present results on more datasets (PASCAL VOC 2012 for object localization, COCO for object segmentation, COCO-Stuff 171 for scene decomposition), additional metrics (detection rate for object localization, per-pixel accuracy for scene decomposition), and list further baseline methods. Finally, we also run the unsupervised semantic segmentation method STEGO on the MOVi datasets (Sec. C.3).

### C.1  UNSUPERVISED OBJECT LOCALIZATION

**Datasets and Metrics**   In the computer vision literature, the task of object localization refers to finding the location of one or more objects that repeatedly appear in a collection of images (Cho et al., 2015). In particular, objects have to be discovered by proposing bounding boxes that "correctly localize" them. An object counts as correctly localized if its bounding box has an IoU-overlap of at least 50% with a proposed bounding box. Like prior work (Vo et al., 2020; Melas-Kyriazi et al., 2022), we report the fraction of images on which at least one object was correctly localized (CorLoc), and the average fraction of objects correctly localized per image (detection rate). As our method produces masks, we use the tightest bounding box around each mask as the proposal. We evaluate on the trainval split of PASCAL VOC 2012 and COCO-20k, as is standard in the literature (Vo et al., 2020; Simeoni et al., 2021). COCO-20k is a subset of COCO 2014 filtered to contain no instances annotated as crowd.

**Results (Table 6)**   The results show that our method manages to focus on the salient parts of the data. On the majority of images, at least one object is captured by a slot. Also, our method reaches results comparable or better to what has been reported in the literature on this task. Interestingly, zero-shot transferring a model trained on COCO to PASCAL yields better results than a model trained on PASCAL itself; we explain this by the size of the training data, which is around $10\times$ larger for COCO.

Table 6: Object Localization (mean ± standard dev., 5 seeds). We report CorLoc and detection rate and compare with rOSD (Vo et al., 2020), DINO-Seg (Simeoni et al., 2021), LOST (Simeoni et al., 2021), DeepSpectral (Melas-Kyriazi et al., 2022) and TokenCut (Wang et al., 2022)

| | PASCAL VOC 2012 | | COCO-20k | |
|---|---|---|---|---|
| | CorLoc | DetRate | CorLoc | DetRate |
| rOSD | 51.9 | 41.2 | 48.5 | 12.0 |
| DINO-Seg | 46.2 | – | 42.1 | – |
| LOST | 64.0 | – | 50.7 | – |
| DeepSpectral | 66.4 | – | 52.2 | – |
| TokenCut | 72.1 | – | 58.8 | – |
| Ours | 69.8 ±4.9 | 52.1 ±3.9 | 67.2 ±1.5 | 31.0 ±0.9 |
| Ours (COCO Transfer) | 77.9 ±1.0 | 58.4 ±0.9 | | |

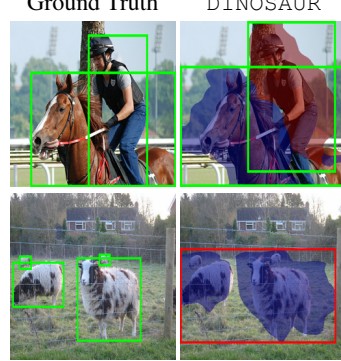

Ground Truth      DINOSAUR

Figure 10: Object Localization on PASCAL VOC 2012.

Table 7: Unsupervised Semantic Segmentation (mean ± standard dev., 5 seeds). We report mean intersection over union (mIoU). For scene decomposition, we additionally report per-pixel accuracy (pAcc). STEGO results are presented without CRF post-processing for fair comparison with other methods. Results marked † were produced ourselves using the official implementation.

(a) Object Segmentation. We compare with MaskContrast (Van Gansbeke et al., 2021), DeepSpectral (Melas-Kyriazi et al., 2022), MaskDistill (Van Gansbeke et al., 2022), and COMUS (Zadaianchuk et al., 2023).

| | PASCAL VOC 2012 | COCO |
|---|---|---|
| MaskContrast | 35.0 | – |
| DeepSpectral | 37.2 | – |
| MaskDistill | 45.8 | – |
| COMUS | 50.0 | 19.6 |
| Ours | 37.2 ±1.8 | 13.9 ±0.4 |

(b) Scene Decomposition. We compare with IIC (Ji et al., 2019), SegDiscover (Huang et al., 2022), PiCIE (Cho et al., 2021), SlotCon (Wen et al., 2022), and STEGO (Hamilton et al., 2022).

| | COCO-Stuff 27 | | COCO-Stuff 171 | |
|---|---|---|---|---|
| | mIoU | pAcc | mIoU | pAcc |
| IIC | 6.7 | 21.8 | – | – |
| SegDiscover | 14.3 | 56.5 | – | – |
| PiCIE + H. | 14.4 | 50.0 | – | – |
| SlotCon | 18.3 | 42.4 | – | – |
| STEGO | 26.8 | 54.8 | 10.0† | 32.5† |
| Ours | 24.0 ±0.9 | 44.9 ±1.2 | 13.1 ±0.3 | 27.0 ±0.3 |

## C.2 Unsupervised Semantic Segmentation

In this task, each pixel of the image has to be assigned a semantic class label. As in the unsupervised setting, no direct correspondence to ground truth classes are given, a successful model needs to output segments that consistently cover the same semantic content on different images.

**Tasks, Metrics and Baselines** We report results on two different versions of this task: in object segmentation, only foreground classes have to segmented; in scene decomposition, the background has to be split into semantic categories as well. For object segmentation, we evaluate on PASCAL VOC 2012 and COCO 2017, with instance masks merged to semantic class masks. For scene decomposition, we evaluate on the COCO-Stuff dataset (Caesar et al., 2018), in two different variants: COCO-27 uses a reduced set of 27 coarse classes from the COCO class hierarchy, whereas COCO-171 uses the full set of 80 foreground and 91 background classes available in the dataset. To the best of our knowledge, we are the first to report results on this challenging setting. We evaluate with mean intersection-over-union over classes (mIoU) and per-pixel accuracy (pAcc).

**Details on Obtaining a Semantic Segmentation with DINOSAUR** For unsupervised semantic segmentation, we need to assign a discrete label to each discovered mask. As our method does not directly output labels required for semantic segmentation, we obtain them by running K-Means clustering on features associated with each slot after training the model, then assigning clusters to ground truth classes by maximizing IoU using Hungarian matching. Similar ways of evaluating are commonly used in the literature (Van Gansbeke et al., 2021; Melas-Kyriazi et al., 2022). In particular, for each slot generated by the model, we compute a feature vector by taking a weighted average between the slot's mask and ViT features (see below). Each vector is then L2-normalized, and the set

of vectors from all images is clustered by running k-means clustering, and each slot is associated with a cluster. Finally, the clusters are assigned to ground truth classes by maximizing the total IoU between clusters and classes using the Hungarian method (Kuhn, 1955). For K-Means clustering, to avoid sensitivity to initialization, we run 20 repetitions and use the run with the lowest sum of squared distances to the cluster centers. As we noticed there is still some variance between different evaluation runs, we run the inference and clustering procedure three times and use the average as the final value for the training run.

*Features for clustering (Table 8).* Which features should we use for clustering the slots? We initially experimented with the slots themselves, but found that they do not cluster well. Instead, we use a weighted average of the slots masks and pre-trained ViT features to produce a feature to cluster for each slot. We compare features from ViT block 9 and ViT block 12 and find that the best block depends on the training method. In the following, we use block 9 for supervised training, DINO and MSN, and block 12 for MoCo-v3 and MAE.

Table 8: Comparing ViT-B/16 blocks and slot features for clustering on COCO-27 (mIoU).

| Method | Slots | Block 9 | Block 12 |
|---|---|---|---|
| Supervised | 8.0 | 18.0 | 13.3 |
| DINO | 13.4 | 24.0 | 21.1 |
| MoCo-v3 | 13.5 | 20.5 | 22.4 |
| MSN | 11.9 | 21.8 | 16.8 |
| MAE | 6.7 | 10.1 | 11.2 |

*Number of clusters.* Results for unsupervised semantic segmentation are dependent on the number of clusters used. Prior methods (Hamilton et al., 2022; Van Gansbeke et al., 2022) typically use the number of classes labeled in the datasets as the number of clusters. For unsupervised object segmentation, this presents an issue for our method, as it captures entities in both foreground and background, and thus clustering using the number of (foreground) object classes leads to over-merging of classes. To ensure a fairer comparison, we instead chose the number of clusters by estimating the how many foreground and background classes can appear on the different datasets. In particular, for PASCAL VOC 2012, we use 105 clusters (the number of classes occurring on at least 0.5% of images, using the PASCAL-Context (Mottaghi et al., 2014) labeling and class statistics), and for COCO, we use 172 clusters (the number of classes labeled on the COCO-Stuff dataset (Caesar et al., 2018)). As this results in more clusters than ground truth classes, Hungarian matching leaves some clusters unassigned; we treat all unassigned clusters as being assigned to the "background" class. For scene decomposition, we simply use 27 clusters for COCO-Stuff 27 and 172 clusters for COCO-Stuff 172.

**Results on Object Segmentation (Table 7a)** This task is not naturally suited for our method, which decomposes the scene into semantically meaningful parts without distinguishing fore- and background. We still obtain competitive results with recently published work (Van Gansbeke et al., 2021; Melas-Kyriazi et al., 2022). Moreover, DeepSpectral, MaskDistill and COMUS employ additional steps of training segmentation networks which improves results and allows them to run at the original image resolution. In contrast, the masks we evaluate are only of size $14 \times 14$; we leave it to future work to improve the resolution of the produced masks.

**Results on Scene Decomposition (Table 7b)** We are competitive with the state-of-the-art method STEGO on the setting with 27 coarse classes. On the novel setting with all 171 classes, our method improves upon STEGO by 3.2 IoU points. This demonstrates that `DINOSAUR` is able to handle the full complexity of real-world scenes somewhat better.

## C.3 Unsupervised Semantic Segmentation Baseline (STEGO) on MOVi

An interesting question is to what degree unsupervised semantic segmentation methods (which we compare to on real-word data in Sec. C.2) can already solve the object discovery task on synthetic datasets designed to benchmark object-centric learning methods. If this were the case, it would showcase that those datasets are flawed in the sense that instance-level grouping (one goal of object-centric learning) is not needed to tackle them. To this end, we test the state-of-the-art scene segmentation method STEGO (Hamilton et al., 2022) on the MOVi datasets.

Table 9: Results for STEGO on the MOVi datasets.

| Dataset | Clusters | ARI | FG-ARI | mBO |
|---|---|---|---|---|
| MOVi-C | 22 | 40.0 | 20.1 | 24.1 |
| | 27 | 39.5 | 19.8 | 23.8 |
| MOVi-E | 22 | 41.1 | 15.6 | 11.5 |
| | 27 | 41.6 | 15.8 | 12.0 |

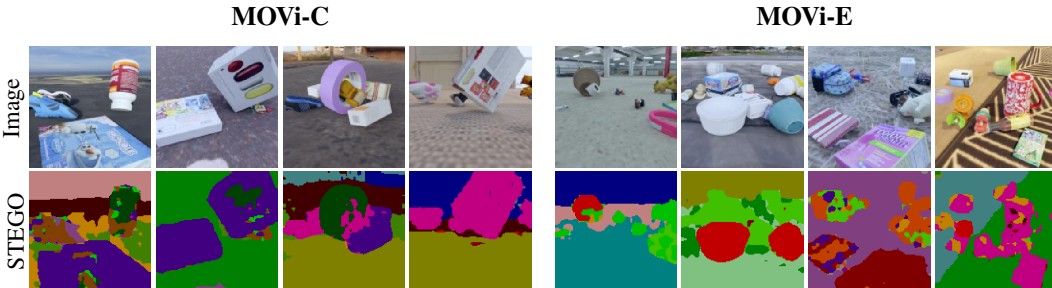

Figure 11: Running the unsupervised semantic segmentation method STEGO (Hamilton et al., 2022) on the MOVi datasets, using 22 clusters for MOVi-C and 27 clusters for MOVi-E. Color codes correspond to the different clusters. Results are presented without CRF post-processing.

STEGO assigns each pixel to one of several clusters/classes, with the clustering performed over the full dataset. We run STEGO with 22 and 27 clusters, corresponding to the 17 semantic categories of the MOVi dataset (see Sec. F), plus 5 or 10 clusters which can be used for the backgrounds. We use a DINO ViT-B/8 backbone for feature extraction, other hyperparameters are listed in Sec. E.2. To evaluate on the object discovery task, we use the segmentation masks produced by STEGO as if they were object masks. That is, each cluster/class mask is used as a separate object mask.

We list quantitative results in Table 9 and show examples in Fig. 11. In addition to foreground ARI (FG-ARI) and mean best overlap (mBO), we list ARI, which takes the background pixels into account as well. STEGO is able to segment objects and background to some degree. However, objects are not properly split, with a single cluster often covering several objects from different semantic categories. Compared to our method, STEGO's masks are also significantly more cluttered and less geometrically consistent. Quantitatively, STEGO performs significantly worse than all object-centric methods. There is a large difference between ARI and FG-ARI, indicating that STEGO is often able to separate out the background, but struggles to separate the objects. For mBO on MOVi-C, STEGO is only 2 points worse than our Slot Attention and STEVE baselines, showing that on scenes with a moderate amount of objects, the masks can capture the objects to some degree (but do not succeed in separating them, see Fig. 11). Overall, this result suggests that a current state-of-the-art unsupervised semantic segmentation method can not satisfactorily solve the object discovery task, justifying the existence of datasets like MOVi to study object-centric learning methods.

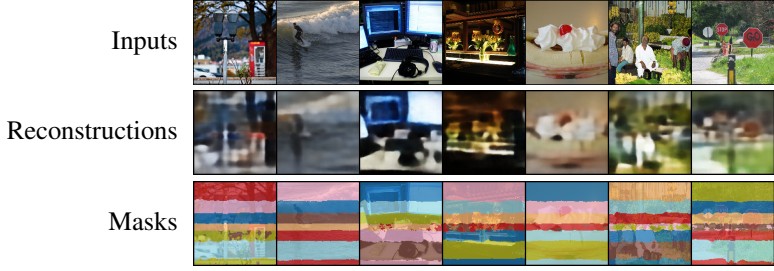

Figure 12: Finetuning a DINO pre-trained ViT encoder using image reconstruction as a training signal fails to yield meaningful results and results in striped mask patterns (see discussion in Sec. 4.3).

# D    ADDITIONAL ANALYSIS

In this section, we conduct several experiments that did not fit into the main part. A brief overview:

Table 10: Results on object localization and semantic segmentation for decoder analysis from Sec. 4.3.

Sec. D.1: Comparing different pre-training schemes for feature extraction and reconstruction.

Sec. D.2: Comparing training a ResNet encoder from scratch vs. using a pre-trained ViT encoder.

Sec. D.3: How does the number of slots affect the quantitative and qualitative results of our method, both at training and test time?

Sec. D.4: How does decoder scale affect the quality of object discovery?

Sec. D.5: Analyzing the poor performance of the Transformer decoder with larger ViTs.

Sec. D.6: Comparing the quality of masks from slot attention and the Transformer decoder.

Sec. D.7: Does DINOSAUR perform *semantic segmentation* on the MOVi datasets?

Table 10: Additional results for decoder analysis in Fig. 3 on *Object Localization* and *Unsupervised Semantic Segmentation* (mean $\pm$ standard dev., 3 seeds), using a ViT-B/16 encoder and 6 slots (PASCAL VOC 2012) or 7 slots (COCO). For localization, results are given on PASCAL VOC 2012 trainval (training on PASCAL VOC 2012) and COCO-20k (training on COCO). For segmentation, results are given for object segmentation (PASCAL VOC 2012, COCO) and scene decomposition (Stuff-27, Stuff-171); when the training dataset is different from the evaluation dataset, this tests zero-shot transfer (i.e. PASCAL $\rightarrow$ COCO and COCO $\rightarrow$ PASCAL). We also refer to Figs. 22 and 23 for side-by-side comparisons of the two types of decoders.

| Training | Decoder | Object Localization | | Semantic Segmentation (mIoU) | | | |
| | | CorLoc | DetRate | COCO | Stuff-27 | Stuff-171 | PASCAL |
| --- | --- | --- | --- | --- | --- | --- | --- |
| PASCAL | MLP | 76.0 $\pm0.3$ | 56.3 $\pm0.2$ | 12.5 $\pm0.4$ | 21.0 $\pm0.3$ | 11.3 $\pm0.1$ | 33.7 $\pm0.9$ |
| | Transformer | 69.8 $\pm4.9$ | 52.1 $\pm3.9$ | 11.9 $\pm1.0$ | 22.2 $\pm0.8$ | 11.6 $\pm0.6$ | 37.2 $\pm1.8$ |
| COCO | MLP | 69.3 $\pm0.5$ | 29.8 $\pm0.3$ | 12.4 $\pm0.2$ | 20.3 $\pm0.2$ | 11.7 $\pm0.0$ | 32.3 $\pm0.8$ |
| | Transformer | 67.2 $\pm1.5$ | 31.0 $\pm0.9$ | 13.9 $\pm0.4$ | 24.0 $\pm0.9$ | 13.1 $\pm0.3$ | 38.6 $\pm1.1$ |

## D.1    CHOICE OF PRE-TRAINING METHOD

We compare different different backbones and pre-training schemes in Table 11. We list results on object discovery, object localization and unsupervised semantic segmentation for a comprehensive overview. Different from Table 2 in the main part, here we consider the setting with pre-trained, frozen encoders. We also investigate target features from ResNet-50 encoders and differently-sized ViTs. When using ResNet50 encoders, we resorted to the MLP decoder as the Transformer decoder did not yield good performance when reconstructing ResNet features.

As in Table 2, we find that all self-supervised schemes perform well for the task of object discovery (cf. Fig. 24). In comparison, supervised training via ImageNet classification is a worse source of features. Moreover, there is a gap between ViT and ResNet features: while ARI is high (likely a by-product of using the MLP decoder), mBO is poor. Visual inspection shows that ResNet-trained models seem to rely on a mix of semantic and geometric grouping of the image (cf. Fig. 25). Generally, larger ViT architectures also yield better results. We conclude from this experiment that scaling the architecture used for the encoder and target features yields better results. Furthermore, we find that using *self-supervised* training is important for real-world grouping to emerge, but that the specific method is less important.

Finally, we also point to Table 12, where we list results for different self-supervised schemes, but training with the MLP decoder instead of the Transformer decoder. This is interesting because MAE, the worst among the self-supervised methods with the Transformer decoder, yields the overall best results in terms of ARI on COCO object discovery: 42.3. This suggests that there are still details about how exactly self-supervised training methods interact with object-centric feature reconstruction left open for investigation.

Table 11: Comparing different pre-trained models for feature extraction *and* reconstruction target (mean $\pm$ standard dev., 3 seeds), using 7 slots, the Transformer decoder for ViTs, and the MLP decoder for ResNets. For discovery, results are given on COCO. For localization, results are given on COCO-20k. For segmentation, results are given for object segmentation (COCO) and scene decomposition (Stuff-27, Stuff-171). We refer to Fig. 24 for side-by-side comparisons.

| Encoder | Pretraining | Object Discovery | | | Object Localization | | Semantic Segmentation (mIoU) | | |
|---|---|---|---|---|---|---|---|---|---|
| | | FG-ARI | mBO$^i$ | mBO$^c$ | CorLoc | DetRate | COCO | Stuff-27 | Stuff-171 |
| ResNet50 | Supervised | 36.0 $\pm$0.3 | 22.1 $\pm$0.1 | 24.2 $\pm$0.1 | 44.2 $\pm$0.6 | 15.0 $\pm$0.3 | 9.1 $\pm$0.1 | 13.7 $\pm$0.3 | 7.7 $\pm$0.1 |
| ResNet50 | DINO | 36.0 $\pm$0.5 | 22.9 $\pm$0.4 | 25.1 $\pm$0.4 | 44.3 $\pm$2.9 | 15.4 $\pm$1.3 | 7.0 $\pm$0.1 | 12.8 $\pm$0.1 | 6.4 $\pm$0.1 |
| ResNet50 | MoCo-v3 | 33.7 $\pm$0.5 | 21.1 $\pm$0.2 | 23.2 $\pm$0.2 | 35.2 $\pm$1.0 | 11.2 $\pm$0.5 | 6.3 $\pm$0.0 | 11.9 $\pm$0.2 | 5.4 $\pm$0.1 |
| ViT-S/16 | Supervised | 26.0 $\pm$0.2 | 24.3 $\pm$0.1 | 30.2 $\pm$0.1 | 56.2 $\pm$0.7 | 24.5 $\pm$0.3 | 18.2 $\pm$0.2 | 14.4 $\pm$0.5 | 13.5 $\pm$0.1 |
| ViT-S/16 | DINO | 36.9 $\pm$0.6 | 29.7 $\pm$0.7 | 33.9 $\pm$1.0 | 68.6 $\pm$1.9 | 30.2 $\pm$1.3 | 11.2 $\pm$0.1 | 20.3 $\pm$0.1 | 10.6 $\pm$0.2 |
| ViT-B/16 | Supervised | 32.3 $\pm$0.8 | 26.3 $\pm$0.3 | 31.5 $\pm$0.3 | 59.1 $\pm$1.1 | 25.5 $\pm$0.4 | 18.6 $\pm$0.2 | 13.3 $\pm$0.2 | 13.5 $\pm$0.1 |
| ViT-B/16 | DINO | 34.1 $\pm$1.0 | 31.6 $\pm$0.7 | 39.7 $\pm$0.9 | 67.2 $\pm$1.5 | 31.1 $\pm$0.9 | 13.9 $\pm$0.4 | 24.0 $\pm$0.9 | 13.1 $\pm$0.3 |
| ViT-B/16 | MoCo-v3 | 35.2 $\pm$0.2 | 31.4 $\pm$0.2 | 38.5 $\pm$0.5 | 70.0 $\pm$0.5 | 32.3 $\pm$0.1 | 14.3 $\pm$0.2 | 22.4 $\pm$0.5 | 13.4 $\pm$0.2 |
| ViT-B/16 | MSN | 35.2 $\pm$0.3 | 31.1 $\pm$0.2 | 37.2 $\pm$0.4 | 69.7 $\pm$0.7 | 31.9 $\pm$0.4 | 13.3 $\pm$0.2 | 21.8 $\pm$0.3 | 12.2 $\pm$0.2 |
| ViT-B/16 | MAE | 32.8 $\pm$3.7 | 30.2 $\pm$1.8 | 33.2 $\pm$1.8 | 68.9 $\pm$5.2 | 29.8 $\pm$3.0 | 9.2 $\pm$0.4 | 11.2 $\pm$0.7 | 7.3 $\pm$0.2 |
| ViT-S/8 | DINO | 34.3 $\pm$0.5 | 32.3 $\pm$0.4 | 38.8 $\pm$0.4 | 73.0 $\pm$0.2 | 33.8 $\pm$0.0 | 13.4 $\pm$0.1 | 23.7 $\pm$0.3 | 13.2 $\pm$0.1 |

## D.2 COMPARISON OF RESNET AND PRE-TRAINED VIT ENCODERS

In this experiment, we compare training a ResNet34 encoder from scratch vs. using a pre-trained, frozen ViT encoder (i.e. our primary setup). As the reconstruction target, we still use features from a pre-trained ViT. Notably, we were not able to optimize the ResNet from scratch using the Transformer decoder, as the model degraded to only use a single slot for the full image. Instead, here we use the MLP decoder with both ResNet34 and pre-trained ViT encoders.

Table 12 lists the results. Example predictions on COCO are included in Table 25 and Table 26. We find that training from scratch is a viable alternative to using a pre-trained encoder, in some cases improving over the pre-trained encoder. We think this is because training from scratch allows adaptation to the domain, whereas the pre-trained encoder may to some degree be misaligned. On COCO, the improvement may also partially stem from the ResNet's higher spatial resolution ($28 \times 28$) — although the decoder's mask resolution is the same ($14 \times 14$). However, using a frozen encoder has computational advantages: no backward pass is needed, and only one forward pass through the encoder is needed (instead of one pass through the encoder being trained, and one pass through the encoder providing the target features). Finally, we find that pre-trained ResNet features are a worse reconstruction target then ViT features. Overall, this experiment shows that the crucial element making our approach work is feature reconstruction. While the type of encoder and the use of pre-trained weights can yield further improvements, they are less important compared to the use of feature reconstruction.

## D.3 NUMBER OF SLOTS ANALYSIS

### D.3.1 TRAINING SENSITIVITY

**MOVi-E (Table 13)** On the synthetic MOVi-E dataset, we found that changing the number of slots can greatly impact performance. We attribute this to the strong impact object splitting (distributing an object over two or more slots) can have on the ARI score. While setting the number of slots to the maximum number of objects that can appear in the scene (23 on MOVi-E) allows the model to fully cover crowded scenes, most scenes have fewer objects, leading to impaired performance on those. We show examples on MOVi-E with 11 slots for all tested methods in Fig. 21.

**PASCAL (Table 14) & COCO (Fig. 14, Fig. 15)** On both PASCAL VOC 2012 and COCO, our method performs well over a range of slots, as long as a minimum number of slots is given. Slot Attention, in contrast, fails to produce meaningful results for all tested number of slots. For our method on COCO, the analysis shows that there are different sweet spots for instance-level (9–11 slots) and class-level segmentation (7 slots). Visual inspection (see Fig. 13) suggests that the optimal number of slots (unsurprisingly) depends on the complexity of the specific image: using few slots tends to lead to grouping of related objects, whereas using many slots leads to oversegmentation of objects into

Table 12: Training a ResNet34 encoder *from scratch* vs. using a *pre-trained frozen* ViT encoder on *object discovery* (mean $\pm$ standard dev., 3 seeds). Both options use the same target features for reconstruction and the MLP decoder. Training from scratch yields overall similar results compared to using a pre-trained encoder. We refer to Figs. 25 and 26 for side-by-side comparisons.

| Dataset | Target Features | | Encoder | Training | FG-ARI | mBO$^{(i,c)}$ | |
|---|---|---|---|---|---|---|---|
| MOVi-C | DINO | ViT-S/8 | ResNet34 | Scratch | 65.1 $\pm$2.1 | 29.1 $\pm$2.0 | |
| | | | ViT-S/8 | Frozen | 67.2 $\pm$0.3 | 38.6 $\pm$0.1 | |
| MOVi-E | DINO | ViT-S/8 | ResNet34 | Scratch | 64.5 $\pm$0.6 | 36.5 $\pm$0.1 | |
| | | | ViT-S/8 | Frozen | 64.7 $\pm$0.7 | 34.1 $\pm$0.1 | |
| PASCAL | DINO | ViT-B/16 | ResNet34 | Scratch | 27.4 $\pm$0.7 | 38.3 $\pm$0.5 | 39.2 $\pm$0.5 |
| | | | ViT-B/16 | Frozen | 24.6 $\pm$0.2 | 39.5 $\pm$0.1 | 40.9 $\pm$0.1 |
| COCO | DINO | ResNet50 | ResNet34 | Scratch | 36.6 $\pm$0.3 | 26.2 $\pm$0.2 | 28.8 $\pm$0.1 |
| | | | ResNet50 | Frozen | 36.0 $\pm$0.5 | 22.9 $\pm$0.4 | 25.1 $\pm$0.4 |
| | | ViT-B/16 | ResNet34 | Scratch | 40.9 $\pm$0.2 | 27.9 $\pm$0.0 | 31.1 $\pm$0.1 |
| | | | ViT-B/16 | Frozen | 40.5 $\pm$0.0 | 27.7 $\pm$0.2 | 30.9 $\pm$0.2 |
| | MoCo-v3 | ResNet50 | ResNet34 | Scratch | 31.9 $\pm$1.1 | 23.9 $\pm$0.5 | 26.7 $\pm$0.4 |
| | | | ResNet50 | Frozen | 33.7 $\pm$0.5 | 21.1 $\pm$0.2 | 23.2 $\pm$0.2 |
| | | ViT-B/16 | ResNet34 | Scratch | 40.4 $\pm$0.6 | 28.1 $\pm$0.2 | 31.1 $\pm$0.1 |
| | | | ViT-B/16 | Frozen | 39.8 $\pm$0.2 | 27.4 $\pm$0.1 | 30.4 $\pm$0.1 |
| | MSN | ViT-B/16 | ResNet34 | Scratch | 40.7 $\pm$0.3 | 27.6 $\pm$0.1 | 30.7 $\pm$0.1 |
| | | | ViT-B/16 | Frozen | 39.0 $\pm$0.1 | 28.1 $\pm$0.0 | 31.6 $\pm$0.1 |
| | MAE | ViT-B/16 | ResNet34 | Scratch | 37.7 $\pm$0.1 | 28.1 $\pm$0.1 | 31.7 $\pm$0.0 |
| | | | ViT-B/16 | Frozen | 42.3 $\pm$0.3 | 29.1 $\pm$0.1 | 32.2 $\pm$0.2 |

parts. Choosing the number of slots per-dataset rather than per-image thus only optimizes results for the average scene. An interesting question for future work is how models could deal with the variance of visual complexity and number of instances that they are faced with in real-world situations.

| 5 slots | 7 slots | 9 slots | 11 slots | 13 slots | 15 slots |
|---|---|---|---|---|---|

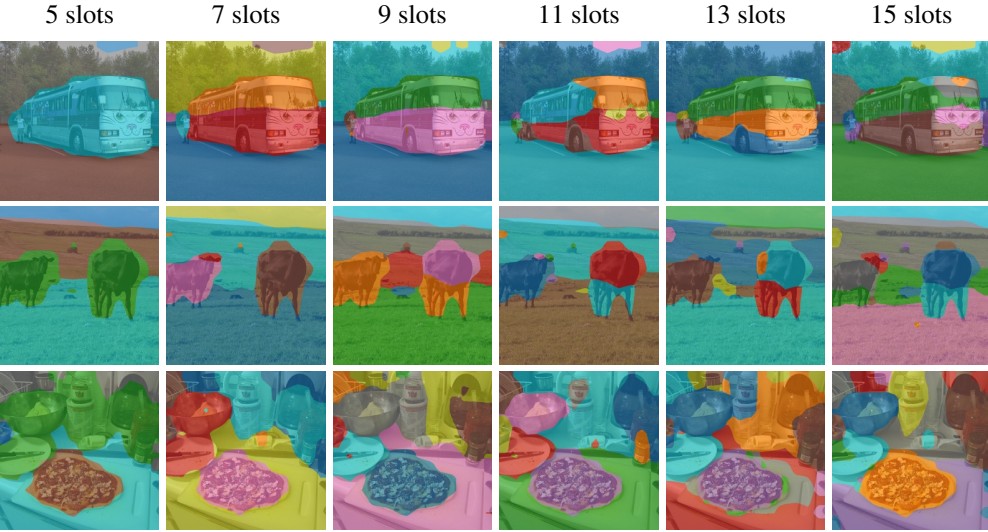

Figure 13: Three scenes of low/medium/high complexity on COCO 2017, varying the number of slots the model is trained with, using a ViT-B/16 encoder and the Transformer decoder. For an appropriate grouping to emerge, the number of slots has to fit the complexity of the scene. However, the groupings found by our method are still semantically meaningful, grouping together related objects (too few slots) or separating objects into their constituents parts (too many slots).

Table 13: Number of slots analysis on *MOVi-E object discovery* (mean over 3 seeds). All methods perform better when training with less slots.

| | 11 slots | | 24 slots | |
|---|---|---|---|---|
| | ARI | mBO | ARI | mBO |
| *Block Masks* | *48.9* | *14.7* | *45.4* | *24.3* |
| *DINO K-Means* | *50.1* | *35.2* | *50.2* | *23.6* |
| Slot Attention | 52.6 | 16.3 | 45.0 | 24.0 |
| SLATE | 49.4 | 16.6 | 44.4 | 23.6 |
| Ours (ViT-S/8) | 76.7 | 29.7 | 64.7 | 34.1 |
| Ours (ViT-B/8) | 79.3 | 32.7 | 65.1 | 35.5 |

Table 14: Number of slots analysis on *PASCAL VOC 2012 object discovery* (mean ± standard dev., 3 seeds), using a ViT-B/16 encoder and the Transformer decoder.

| Slots | FG-ARI | $mBO^i$ | $mBO^c$ |
|---|---|---|---|
| 4 | 29.6 ±1.4 | 34.2 ±1.0 | 40.4 ±1.1 |
| 5 | 26.6 ±1.1 | 39.1 ±0.6 | 45.9 ±0.7 |
| 6 | 24.8 ±2.2 | 44.0 ±1.9 | 51.2 ±1.9 |
| 7 | 19.1 ±1.3 | 42.7 ±0.7 | 47.9 ±0.6 |
| 9 | 20.5 ±0.7 | 43.5 ±0.3 | 47.4 ±0.4 |

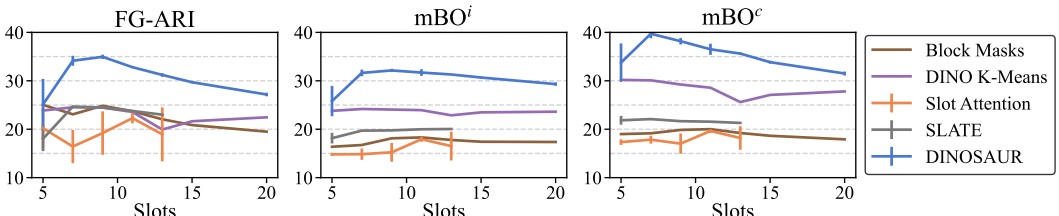

Figure 14: Varying the number of slots the model is trained with on *COCO object discovery* (mean ± standard dev., 3 seeds), using a ViT-B/16 encoder and the Transformer decoder. Our method shows a small trade-off between instance- (more slots) and class-level segmentation (less slots). Slot attention fails to produce meaningful masks over a range of slots.

### D.3.2 TEST-TIME ROBUSTNESS

In the last section, we found that the quality of results is affected by the number of slots the model is trained with. An attractive quality of Slot Attention-based models[3] is that the number of slots *can be changed after the model is trained* (Locatello et al., 2020), and we inherit this feature. However, this constitutes a form of distribution shift for the model, and so it is important to know how robust the model is to such changes. To this end, we compare models on trained on a certain number of slots, with models trained on a *different* number of slots, but tested with that number of slots. If the performance is similar, we have evidence for the robustness of the model to this type of shift.

---

[3]This is only true when using randomly sampled, and not fixed, learned slot initializations. Recent models (Kipf et al., 2022; Elsayed et al., 2022) depart from the original random sampling design, as it appears to be easier to train the model with fixed initializations on more complex data. We found no such difficulties with `DINOSAUR`.

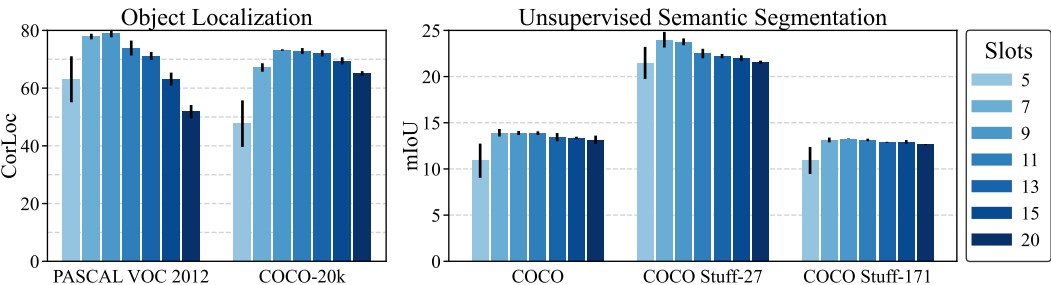

Figure 15: Varying the number of slots the model is trained with on *object localization* and *unsupervised semantic segmentation* (mean ± standard dev., 3 seeds), using a ViT-B/16 encoder and the Transformer decoder. The models are trained on COCO and tested on the respective datasets.

Table 16 shows the results of this experiment on MOVi-E, PASCAL VOC 2012, and COCO. We find that the model is fairly robust to changing the number of slots at test time. Interestingly, in many instances, there is even a large improvement from changing the number of slots (e.g. on MOVi-E from 11 to 24 slots, or on PASCAL, from 6 to 4 slots. This can be explained by our observations in Sec. D.3.1: depending on the number of slots at training time, the model learns to focus on object of different scales. Some object scales fit better to the dataset annotations than others. Applying a better model with a different number of slots can thus improve over models that did not perform well with the original slot configuration. We can also explain the cases where performance drops from changing the number of slots (e.g. on MOVi-E from 24 to 11 slots, or on COCO from 20 slots) with this: those configurations did not work well in the first place, so they also do not transfer well to a different number of slots. Overall, this result shows that `DINOSAUR` generalizes well when applied with a different number of slots.

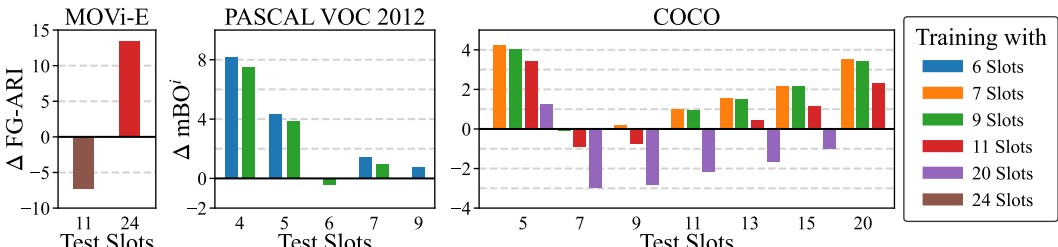

Figure 16: Varying the *test time* number of slots at on *object discovery* (mean $\pm$ standard dev., 3 seeds). We plot the *difference* between testing at a certain number of slots and directly training at that number of slots, for models trained with different number of slots. We list FG-ARI for MOVi-E, and mBO$^i$ for COCO and PASCAL VOC 2012. For MOVi-E, we use a ViT-B/8 encoder with the MLP decoder. For COCO and PASCAL VOC 2012, we use a ViT-B/16 encoder with the Transformer decoder. The performance of the original models can be gauged from Table 13, Table 14 and Fig. 14.

## D.4 How Does Decoder Scale Interact With Object Discovery?

Extending our decoder analysis from Sec. 4.3, we investigate how the capacity of the decoder and the slot dimensionality affects the object discovery performance on COCO. In particular, we compare the MLP with the Transformer decoder: for the MLP decoder, we use a 4-layer MLP and test hidden layer dimensionalities 1024, 2048, and 4096. For the Transformer decoder, we test different layer numbers 2, 3, and 4. For each decoder setting, we test it under slot dimensionalities 64, 128, 256, and 512.

We show the results in Fig. 17. The general trend of Table 3 stays unchanged: the MLP decoder has higher ARI (indicating better instance grouping), whereas the Transformer decoder has higher mBO (indicating better mask quality). For the MLP decoder, we find that scaling the MLP decoder has slight benefits in terms of ARI, and that it is robust to the slot dimensionality (with the exception of slot dimensionality 512). For the Transformer decoder, scaling appears to have little impact. However, we find that the Transformer decoder is sensitive to the slot dimensionality: setting it too low (64, 128) can cause the model to collapse to just use one or two slots to model the whole image. Setting it too high (512) causes training instabilities (i.e. NaNs or loss spikes). These instabilities under slot dimensionality 512 also occur with the MLP decoder. We conjecture this might be related to the use of 16-bit mixed precision and the attention mechanism of slot attention, but we did not investigate further.

## D.5 The Transformer Decoder Performs Poor with Larger ViTs

In Table 15, we compare the Transformer decoder with different ViT encoders on COCO object discovery. We find that scaling up the encoder leads to better results. However, specifically combining the Transformer decoder with the ViT-B/8 encoder leads to poor results – the MLP decoder has no such issues (cf. the results on the MOVi datasets in Fig. 3). Visual inspection shows that the model collapses to use just one or two slots for the whole image. We attribute this to the *high capacity of the decoder*, which leads to it being able to reconstruct the token map well while using the conditioning information from only a few slots. Note that the capacity of the decoder is related to the encoder through the token dimensionality, and the number of tokens (which has strong influence on a

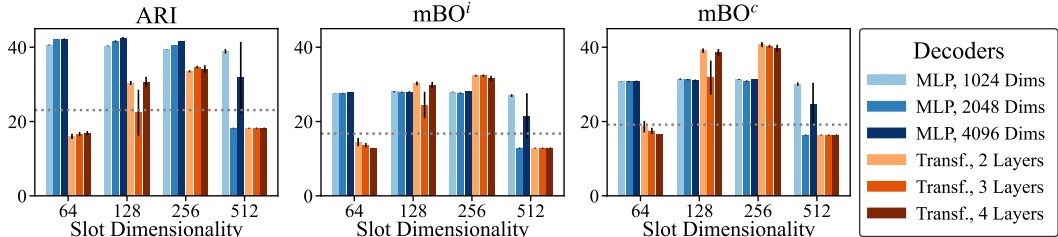

Figure 17: *COCO object discovery* performance when varying decoder capacity and slot dimensionality, using 7 slots (mean ± standard dev., 3 seeds). For the MLP decoder, we vary the dimensionality of a 4-layer MLP, whereas for the Transformer decoder, we vary the number of layers. The dotted line represents the block masks baseline. The analysis reveals instabilities of the Transformer decoder: for too small or too large slot dimensionalities, the performance degrades below the block masks baseline. The MLP decoder is generally robust, but also becomes unstable at slot dimensionality 512.

Transformer's capacity as it directly influences the amount of memory and computation available throughout the blocks). Indeed, when reducing the decoder's capacity by either reducing the feature dimensionality (ViT-S/8) or the number of tokens the decoder operates on (using bicubic interpolation on the feature map to reconstruct), the problem disappears. In principle, the issue could also arise from optimization becoming unstable due to switching to a larger model, but the loss curves do not suggest such instabilities.

Table 15: Comparing different encoders with the Transformer decoder on *COCO object discovery*. For ViT-B/8, we also list results for *scaling the target token map* by a factor of $0.5$ (from 784 to 196 tokens) and by a factor of $0.75$ (to 441 tokens). Object-centricness does no longer emerge for ViT-B/8, which we attribute to the high capacity of the decoder: when reducing the capacity (by reducing the feature dimensionality or the number of tokens), the problems disappear.

| Encoder | Target Tokens | $D_{\text{feat}}$ | FG-ARI | mBO$^i$ | mBO$^c$ |
|---|---|---|---|---|---|
| ViT-S/16 | 196 | 384 | 36.9 ±0.6 | 29.7 ±0.7 | 33.9 ±1.0 |
| ViT-B/16 | 196 | 768 | 34.1 ±1.0 | 31.6 ±0.7 | 39.7 ±0.9 |
| ViT-S/8 | 784 | 384 | 34.3 ±0.5 | 32.3 ±0.4 | 38.8 ±0.4 |
| ViT-B/8 | 196 (scaled) | 768 | 35.8 ±0.7 | 32.0 ±0.6 | 38.6 ±0.2 |
| ViT-B/8 | 441 (scaled) | 768 | 20.1 ±9.8 | 19.1 ±8.6 | 23.5 ±9.9 |
| ViT-B/8 | 784 | 768 | 13.1 ±3.5 | 16.6 ±3.9 | 21.3 ±5.1 |

### D.6  MASK TYPE ANALYSIS

As noted in Sec. 3, the Transformer decoder offers a choice between attention masks from the slot attention module, and attention masks from the decoder. In Table 16, we compare the two types of masks and find that decoder attention masks are consistently better on all tasks, and especially outperform the masks from slot attention on segmentation. Note that we could in principle also use masks from slot attention with the MLP decoder. We point to Singh et al. (2022b), who analyze this choice and find that alpha masks produced by reconstruction are superior to attention masks.

Table 16: Comparing masks from Slot Attention and the Transformer decoder on *COCO* (mean ± standard dev., 5 seeds), using a ViT-B/16 encoder. For segmentation, we list results for object segmentation (COCO) and scene decomposition (Stuff-27, Stuff-171).

| Mask Type | Object Discovery | | | Object Localization | | Segmentation (mIoU) | | |
|---|---|---|---|---|---|---|---|---|
| | FG-ARI | mBO$^i$ | mBO$^c$ | CorLoc | DetRate | COCO | Stuff-27 | Stuff-171 |
| Slot Attention | 31.7 ±1.4 | 30.3 ±0.8 | 38.1 ±1.1 | 66.0 ±1.5 | 30.2 ±0.8 | 12.0 ±0.2 | 21.1 ±0.6 | 11.7 ±0.2 |
| Decoder | 34.1 ±1.0 | 31.6 ±0.7 | 39.7 ±0.9 | 67.2 ±1.5 | 31.1 ±0.9 | 13.9 ±0.4 | 24.0 ±0.9 | 13.1 ±0.3 |

### D.7 Does DINOSAUR Perform Semantic Grouping on MOVi?

On the MOVi datasets, `DINOSAUR` seemingly separates objects into slots, i.e. instance grouping. As `DINOSAUR` exhibits some bias towards semantic grouping on real-world datasets, an interesting question is whether our approach does in fact also perform semantic grouping on MOVi. If this would be the case, it would be hard to detect by evaluating on the original MOVi datasets, because objects of the same category rarely appear on the same image on those datasets.

To answer this question, we designed two variants of the MOVi-C dataset. In the first, only objects of *the same semantic category* appear in each image. In the second, each image contains *the exact same object* several times. All other characteristics of the dataset stay unchanged. If our method would perform semantic grouping on MOVi, objects would be grouped together on those datasets. In Table 17, we list the results for training on MOVi-C and testing on the new "semantic" variants. We find that the results do not change significantly compared to evaluating on the original MOVi-C dataset, indicating that our method does in fact perform instance grouping on the MOVi datasets.

Table 17: Testing a model trained on MOVi-C on two variants of MOVi-C: in "semantic objects", each image only contains objects of the same semantic category; in "duplicate objects", each image only contains the same object, but several times. We show results for our method with ViT-B/8 encoder and MLP decoder, with mean $\pm$ standard dev. over 5 seeds. We find that the results do not change significantly, indicating that `DINOSAUR` does not apply semantic grouping on MOVi.

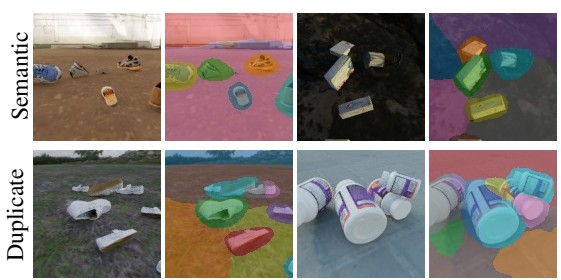

| Dataset | FG-ARI | mBO |
|---|---|---|
| MOVi-C | 68.6 $_{\pm 0.4}$ | 40.8 $_{\pm 0.2}$ |
| Semantic Objects | 68.6 $_{\pm 0.4}$ | 39.1 $_{\pm 0.2}$ |
| Duplicate Objects | 68.1 $_{\pm 0.2}$ | 39.0 $_{\pm 0.2}$ |

## E Implementation Details

### E.1 Architecture and Hyperparameters

In the following, we describe implementation details of the `DINOSAUR` architecture. The hyperparameters for our experiments are given in Table 18.

**ViT Encoder** We use the Vision Transformer implementation and provided by the timm library (Wightman, 2019)[4]. Depending on the experiment, we use the following configurations: ViT-S/16 (token dimensionality 384, 6 heads, patch size 16), ViT-S/8 (token dimensionality 384, 6 heads, patch size 8), ViT-B/16 (token dimensionality 768, 12 heads, patch size 16), ViT-B/8 (token dimensionality 768, 12 heads, patch size 8). The specific timm model names (for DINO weights) used are `vit_small_patch16_224_dino` (ViT-S/16), `vit_small_patch8_224_dino` (ViT-S/8), `vit_base_patch16_224_dino` (ViT-B/16), `vit_base_patch8_224_dino` (ViT-B/8). All models use 12 Transformer blocks, linear patch embedding and additive positional encoding. The output of the last block (not applying the final layer norm) is passed on to the Slot Attention module and used in the feature reconstruction loss, after removing the entry corresponding to the CLS token. We always use images of size $224 \times 224$ as input to the ViT; this yields $N = 14^2 = 196$ patches for patch size 16 and $N = 28^2 = 784$ patches for patch size 8. If the image size is not equal to 224, we use bicubic interpolation for resizing. For the pre-trained weights, we use the timm library for DINO and third-party releases for MoCo-v3[5], MSN[6], and MAE[7].

---

[4] https://github.com/rwightman/pytorch-image-models, v0.6.7
[5] https://github.com/facebookresearch/moco-v3
[6] https://github.com/facebookresearch/msn
[7] https://github.com/facebookresearch/mae

**ResNet34 Encoder**    We use a ResNet34 using the modifications suggested by Kipf et al. (2022) and Elsayed et al. (2022). That is, we modify the basic ResNet34 architecture (`resnet34` in the timm library) to use a stride of 1 in the first convolution, and replace batch normalization with group normalization. This results in a 512-dimensional feature map of size $28 \times 28$ for images of size $224 \times 224$. Like Kipf et al. (2022), we add a linear positional encoding to the feature map passed to the Slot Attention module (coordinates in the four cardinal directions scaled to $[-1, 1]$, linearly projected to the feature dimension and added to the features).

**Slot Attention**    The Slot Attention module largely follows the original formulation from Locatello et al. (2020). After applying a layer norm, the set of $N$ input features is transformed to slot dimensionality $D_{\text{slots}}$ by a two-layer MLP with hidden size equal to the feature size $D_{\text{feat}}$, followed by another layer norm. An initial set of $K$ slots is randomly sampled from a normal distribution parametrized by learnable mean $\mu \in \mathbb{R}^{D_{\text{slots}}}$ and log standard deviation $\log \sigma \in \mathbb{R}^{D_{\text{slots}}}$. The slots are then updated over several iterations, where each iteration involves a layer norm step on the current slots, an attention step where slots compete for input features, updating the slots using a GRU (Cho et al., 2014), and applying a residual two-layer MLP. The query/key/value projections' size is the same as the slot dimensionality $D_{\text{slots}}$; they do not use a bias. The key-value dot product is scaled by $D_{\text{slots}}^{-0.5}$, and an epsilon of $10^{-8}$ is added to the softmax weights for stability reasons. The MLP has a hidden size equal to $4 \cdot D_{\text{slots}}$.

On MOVi-C and MOVi-E, we use 11 and 24 slots respectively, which corresponds to the maximal number of objects that can appear in the scene plus one slot that can capture the background. On PASCAL VOC and COCO, we use 6 and 7 slots respectively, as we found those values to perform quantitatively well and give visually appealing results (cf. App. D.3).

**MLP Decoder**    We use a four-layer MLP with ReLU activations, with output dimensionality $D_{\text{feat}} + 1$, where the last dimension is used for the alpha value. The MLP has hidden layer sizes of 1024 for the MOVi datasets, and 2048 for COCO. A learnable positional encoding of size $N \times D_{\text{slots}}$ is added to the slots after broadcasting them to the number of patches $N$.

**Transformer Decoder**    We use the standard decoder design from Vaswani et al. (2017) with pre-normalization (Xiong et al., 2020), conditioning on the set of slots output by the slot attention module. Similar to ImageGPT (Chen et al., 2020a), the decoder autoregressively reconstructs $N$ patch features of dimensionality $D_{\text{feat}}$ starting from the top-left and proceeding row-by-row over the image ("raster order"). To this end, its initial input is the set of target features shifted back by one, removing the last feature and inserting a learned "beginning-of-sentence" feature at the start. Each Transformer block then consists of self-attention on the set of tokens (using causal masking to prevent attending to future tokens), cross-attention with the set of slots, and a residual two-layer MLP with hidden size $4 \cdot D_{\text{feat}}$. Before the Transformer blocks, both the initial input and the slots are linearly transformed to $D_{\text{feat}}$, followed by a layer norm. The Transformer thus operates on tokens of dimensionality $D_{\text{feat}}$. The output of the last Transformer block is directly used as the reconstruction $\boldsymbol{y}$. We do not apply dropout as we did not find it to be beneficial. Also, we do not add positional encoding as the target features forming the input already contain positional information (from the ViT encoder).

**Masks for Evaluation**    As mentioned in the main text, we use the attention mask from the decoder as the slot mask used for evaluation. In particular, the cross attention step involves a softmax over the slots for each of the $N$ tokens, resulting in $N$ attention weights per slot, which can be reshaped to form a $\sqrt{N} \times \sqrt{N}$ mask. Intuitively, each mask entry measures how important the corresponding slot was for reconstructing the patch feature at that position. For the evaluation, this mask is bilinearly resized to the image size. We use the attention mask from the last decoder block, which we found to perform best against the alternatives of using masks from earlier blocks or averaging the masks across blocks. The soft slot masks are converted to hard masks using the $\arg\max$.

**Details for Specific Experiments**    Here we list settings for experiments aside from our main experiments.

- Insufficiency of Image Reconstruction (Sec. 4.3). In this experiment, we train on COCO using image reconstruction and combine a ViT-B/16 encoder with Slot Attention's spatial broadcast decoder (as detailed in App. E.2). Image reconstruction uses $128 \times 128$ as the target resolution. For finetuning, we use a lower learning rate of $4 \cdot 10^{-5}$ for the encoder, compared to $4 \cdot 10^{-4}$ for Slot Attention and the decoder. All models use 7 slots. Other training and architecture settings are the same as for the main experiments.

- Choice of Self-Supervised Method (Sec. 4.3) and Choice of Pre-Training Method (Sec. D.1). In these experiments, we use different pre-trained networks for feature extraction and reconstruction. The specific timm model names used are `resnet50` (ResNet50, supervised), `vit_small_patch16_224` (ViT-S/16, supervised), `vit_base_patch16_224` (ViT-B/16, supervised), `vit_small_patch16_224_dino` (ViT-S/16, DINO), `vit_base_patch16_224_dino` (ViT-B/16, DINO). For MoCo-v3, MSN and MAE, we use the ViT-B/16 implementation of timm with third-party weight releases (referenced above). All networks were pre-trained on the ImageNet dataset. The ResNet50 encoder uses the features output from the second last block (feature level 3 in timm), resulting in a 1024-dimensional feature map of size $14 \times 14$. Like Kipf et al. (2022), we add linear positional encoding to this feature map before Slot Attention. The ViT encoders use the Transformer decoder whereas for the ResNet encoder, we use the MLP decoder as training with the Transformer decoder did not yield good results. All models use 7 slots.

- Choice of Decoder (Sec. 4.3). This experiment uses 11 slots for MOVi-C, 6 slots for PASCAL VOC 2012, and 7 slots for COCO.

- Object Property Prediction (Sec. B.3). We train the downstream models for 15 000 steps with the Adam optimizer with a starting learning rate of $10^{-3}$ and batch size 64. The learning rate is halved every 2 000 training steps. The non-linear predictor is a one-hidden layer MLP with 256 hidden neurons and leaky ReLU activation (slope 0.01). Continuous targets are standardized by computing mean and variance over the training dataset. We use center crops as input and filter objects that are not contained in the center crop.

- Comparison of ResNet and Pre-Trained ViT Encoders (Sec. D.2). We train the ResNet models for 300k steps (instead of 500k steps as the models in our main experiments) due to computational constraints, but we found that they can also benefit from longer training.

Table 18: Hyperparameters for `DINOSAUR` used for the results on MOVi-C, MOVi-E, PASCAL VOC 2012, COCO and KITTI datasets.

| Dataset | | **MOVi-C** | **MOVi-E** | **PASCAL** | **COCO** | **KITTI** |
|---|---|---|---|---|---|---|
| Training Steps | | 500k | 500k | 250k | 500k | 500k |
| Batch Size | | 64 | 64 | 64 | 64 | 64 |
| LR Warmup Steps | | 10000 | 10000 | 10000 | 10000 | 10000 |
| Peak LR | | 0.0004 | 0.0004 | 0.0004 | 0.0004 | 0.0004 |
| Exp. Decay Half-Life | | 100k | 100k | 100k | 100k | 100k |
| ViT Architecture | | ViT-B | ViT-B | ViT-B | ViT-B | ViT-B |
| Patch Size | | 8 | 8 | 16 | 16 | 8 |
| Feature Dim. $D_{\text{feat}}$ | | 768 | 768 | 768 | 768 | 768 |
| Gradient Norm Clipping | | 1.0 | 1.0 | 1.0 | 1.0 | 1.0 |
| Image/Crop Size | | 224 | 224 | 224 | 224 | 224 |
| Cropping Strategy | | Full | Full | Random | Center | Random |
| Augmentations | | – | – | Random Horizontal Flip | | |
| Image Tokens | | 784 | 784 | 196 | 196 | 784 |
| Decoder | Type | MLP | MLP | Transformer | Transformer | MLP |
| | Layers | 4 | 4 | 4 | 4 | 4 |
| | Heads | – | – | 8 | 8 | – |
| | MLP Hidden Dim. | 1024 | 1024 | 3072 | 3072 | 2048 |
| Slot Attention | Slots | 11 | 24 | 6 | 7 | 9 |
| | Iterations | 3 | 3 | 3 | 3 | 3 |
| | Slot Dim. $D_{\text{slots}}$ | 128 | 128 | 256 | 256 | 32 |
| | MLP Hidden Dim. | 512 | 512 | 1024 | 1024 | 128 |

## E.2 BASELINES

**Block Masks**   The block mask patterns are generated by equally dividing the image into a specified number of columns, then equally dividing the columns such that a target number of slot masks is reached (see Fig. 18). For less than 9 masks, 2 columns are used, for 9–15 masks, 3 columns, and for more than 15 masks, 4 columns are used. For the block masks results given in the main text, MOVi-C uses 11 masks, MOVi-E uses 24 masks, PASCAL VOC 2012 uses 6 masks and COCO uses 7 masks. The number of masks is consistent with the number of slots used by the other methods on the respective datasets.

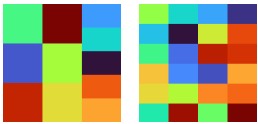

Figure 18: Block masks for 11 and 24 slots.

**DINO K-Means**   We run the K-Means algorithm on the patch tokens output by the last DINO ViT block (before the final layer normalization). For the MOVi experiments, we use ViT-B/8, whereas for the PASCAL VOC 2012 and COCO experiments, we use ViT-B/16. The resulting low-dimensional clustering masks are resized to the target mask resolution required for evaluation, where we treat each cluster as an object instance. For K-Means, we use the standard scikit-learn implementation (*sklearn.cluster.KMeans*) with default parameters (10 repetitions, k-means++ init, maximum of 300 iterations). We only vary the number of clusters: for the results given in the main text, MOVi-C uses 11 clusters, MOVi-E uses 24 clusters, PASCAL VOC 2012 uses 6 clusters and COCO uses 7 clusters. The number of clusters is consistent with the number of slots used by the other methods on the respective datasets.

**Slot Attention**   The hyperparameters used to run the experiments can be found in Table 19. For the encoder, we use the ResNet34 as described above. The slot attention module largely follows the description given in App. E.1 for the DINOSAUR architecture. However, like Kipf et al. (2022), we add a linear positional encoding to the feature map (coordinates in the four cardinal directions scaled to $[-1, 1]$, linearly projected to the feature dimension and added to the features). The number of slots was set to 11 and 24 for MOVi-C and MOVi-E, whereas for PASCAL and COCO we used 6 and 7 slots, to be consistent with DINOSAUR. For PASCAL and COCO, we verified that Slot Attention does not succeeded in successfully discovering objects over a range of number of slots (cf. Fig. 14). For the decoder, we use the same spatial broadcast decoder as (Kipf et al., 2022), i.e. 5 transposed convolutions with $5 \times 5$ kernels, channel size 64 and ReLU activations, upsampling from size $8 \times 8$ to $128 \times 128$. Input and target images are scaled to range $[-1, 1]$.

Table 19: Hyperparameters used for Slot Attention training on MOVi-C, MOVi-E, PASCAL VOC 2012 and COCO datasets.

| Dataset | | **MOVi-C** | **MOVi-E** | **PASCAL VOC 2012 & COCO** |
|---|---|---|---|---|
| Training Steps | | 250k | 250k | 250k |
| Batch Size | | 64 | 64 | 64 |
| LR Warmup Steps | | 2500 | 2500 | 2500 |
| Peak LR | | 0.0002 | 0.0002 | 0.0002 |
| Annealing | | Cosine | Cosine | Cosine |
| Gradient Norm Clipping | | 1.0 | 1.0 | 1.0 |
| Image/Crop Size | | 128 | 128 | 128 |
| Cropping Strategy | | Full | Full | Random (PASCAL) / Center (COCO) |
| Augmentations | | – | – | Random Horizontal Flip |
| Slot Attention | Slots | 11 | 24 | 6 & 7 |
| | Iterations | 3 | 3 | 3 |
| | Slot Dim. | 256 | 256 | 256 |
| | MLP Hidden Dim. | 512 | 512 | 512 |

**SLATE**   We use the official SLATE implementation[8]. The hyperparameters are close to the ones used in SLATE (Singh et al., 2022a) and STEVE (Singh et al., 2022b) and are presented in Table 20.

---

[8] https://github.com/singhgautam/slate

We noticed that SLATE's performance metrics began to degrade significantly after some training time; we picked the training time such that best performance was obtained.

Table 20: Hyperparameters used for SLATE training on MOVi-C, MOVi-E, PASCAL VOC 2012 and COCO datasets.

| Dataset | | MOVi-C | MOVi-E | PASCAL VOC 2012 & COCO |
|---|---|---|---|---|
| Training Steps | | 40k | 40k | 50k & 40k |
| Batch Size | | 64 | 64 | 64 |
| LR Warmup Steps | | 30000 | 30000 | 30000 |
| Peak LR | | 0.0001 | 0.0001 | 0.0001 |
| Dropout | | 0.1 | 0.1 | 0.1 |
| Gradient Norm Clipping | | 1.0 | 1.0 | 1.0 |
| DVAE | Vocabulary Size | 4096 | 4096 | 4096 |
| | Temp. Cooldown | 1.0 to 0.1 | 1.0 to 0.1 | 1.0 to 0.1 |
| | Temp. Cooldown Steps | 30000 | 30000 | 30000 |
| | LR | 0.0003 | 0.0003 | 0.0003 |
| Image Size | | 128 | 128 | 128 |
| Image Tokens | | 1024 | 1024 | 1024 |
| Transformer Decoder | Layers | 8 | 8 | 8 |
| | Heads | 4 | 4 | 4 |
| | Hidden Dim. | 192 | 192 | 192 |
| Slot Attention | Slots | 11 | 24 | 6 & 7 |
| | Iterations | 3 | 3 | 3 |
| | Slot Dim. | 192 | 192 | 192 |

**STEVE**   We use the official STEVE implementation[9]. We use the proposed configuration for the MOVi datasets and only change the number of slots to 11 (for MOVi-C) and 24 (for MOVi-E). We train the model for 20 000 steps.

**SAVi++**   We use the official SAVi++ implementation[10]. We use the proposed configuration for MOVi-E and only change it to use fixed, learned slots instead of first-frame conditioning, and train the model with batch size 32 for 30 000 steps. We train with both image reconstruction and optical flow reconstruction.

**STEGO**   To evaluate STEGO on the COCO-Stuff 171 dataset (Sec. C.2), we modified the official STEGO implementation[11] to use batch k-means clustering with 172 clusters. Other parameters were not changed from the STEGO configuration on the COCO-Stuff 27 dataset as the training data is the same for those two datasets.

For the experiment evaluating STEGO on the MOVi datasets (Sec. C.3), we run a DINO ViT-B with patch size 8, using 22 and 27 clusters, and train for 20 000 steps. We manually tested different hyperparameters and settled on $\lambda_{\text{self}} = 0.6$, $\lambda_{\text{knn}} = 0.3$, $\lambda_{\text{rand}} = 0.6$, $b_{\text{self}} = 0.26$, $b_{\text{knn}} = 0.21$, $b_{\text{rand}} = 0.4$ for MOVi-C, and $\lambda_{\text{self}} = 0.6$, $\lambda_{\text{knn}} = 0.3$, $\lambda_{\text{rand}} = 0.6$, $b_{\text{self}} = 0.22$, $b_{\text{knn}} = 0.17$, $b_{\text{rand}} = 0.4$ for MOVi-E. We refer to Hamilton et al. (2022) for the meaning of those parameters.

# F   DATASET AND EVALUATION SETTINGS

In this section, we provide details about the datasets, evaluation settings and metrics used in this work. An overview over which datasets are used for which task can be found in Table 21.

**MOVi**   As the MOVi datasets are video datasets, we convert them into an image dataset by sampling 9 random frames per clip. This yields 87 633 training images for MOVi-C and 87 741 images on

---

[9] https://github.com/singhgautam/steve
[10] https://github.com/google-research/slot-attention-video/
[11] https://github.com/mhamilton723/STEGO

Table 21: Overview over tasks and datasets used in this work. For training, we only use images of the respective datasets, and no labels. For central crops, we first resize the mask such that the short side is 320 pixels long, then take the most centered crop of size $320 \times 320$.

| Dataset | Images | Description | Citation |
|---|---|---|---|
| MOVi-C | 87 633 | Train split w. videos | Greff et al. (2022) |
| MOVi-C validation | 6 000 | Val. split w. instance segm. labels | Greff et al. (2022) |
| MOVi-E | 87 741 | Train split w. images | Greff et al. (2022) |
| MOVi-E validation | 6 000 | Val. split w. instance segm. labels | Greff et al. (2022) |
| VOC 2012 validation | 1 449 | Val. split w. instance segm. labels | Everingham et al. (2012) |
| VOC 2012 trainaug | 10 582 | Train split w. instance segm. labels | Everingham et al. (2012) and Hariharan et al. (2011) |
| VOC 2012 trainval | 11 540 | Detection train and val. splits | Everingham et al. (2012) |
| COCO 2017 | 118 287 | Train split w. instance segm. labels | Lin et al. (2014) |
| COCO 2017 validation | 5 000 | Val split w. instance segm. labels | Lin et al. (2014) |
| COCO-20k | 20 000 | COCO 2014 train& val subset w. det. labels | e.g. Simeoni et al. (2021) |
| COCO-27 | 5 000 | Segm. labels from COCO-Stuff | Caesar et al. (2018), e.g. Ji et al. (2019) |
| COCO-171 | 5 000 | Segm. labels from COCO-Stuff | Caesar et al. (2018), new |
| KITTI | | Raw videos | Geiger et al. (2013) |
| KITTI segmentation | 200 | Images w. instance segm. labels | Alhaija et al. (2018) |

| Task | Training Dataset | Eval Dataset | Eval Resolution | Crop Type |
|---|---|---|---|---|
| Object Discovery | MOVi-C | MOVi-C validation | $128 \times 128$ | Full |
| | MOVi-E | MOVi-E validation | $128 \times 128$ | Full |
| | VOC 2012 trainaug | VOC 2012 validation | $320 \times 320$ | Central |
| | COCO 2017 | COCO 2017 validation | $320 \times 320$ | Central |
| | KITTI | KITTI segmentation | $1242 \times 375$ | Full |
| Object Localization | VOC 2012 trainaug | VOC 2012 trainval | Original | Full |
| | COCO 2017 | COCO-20k | Original | Full |
| Object Segmentation | VOC 2012 trainaug | VOC 2012 validation | Original | Full |
| | COCO 2017 | COCO 2017 validation | Original | Full |
| Scene Segmentation | COCO 2017 | COCO-27 | $320 \times 320$ | Central |
| | COCO 2017 | COCO-171 | $320 \times 320$ | Central |

MOVi-E. For evaluation, we use all frames from the 250 clips, yielding 6 000 images. Note that we use the validation split provided by the MOVi dataset for evaluation, and not the test split containing out-of-distribution objects and backgrounds. This is in line with prior work (Kipf et al., 2022; Elsayed et al., 2022).

For the "semantic" MOVi-C variants used in Sec. D.7, we generated 250 videos using the Kubric simulator (Greff et al., 2022), yielding 2 250 images. For the "semantic objects" variant, after sampling the first object in each video, only objects of the same category were allowed to be sampled. The categories are: "Action Figures", "Bag", "Board Games", "Bottles and Cans and Cups", "Camera", "Car Seat", "Consumer Goods", "Hat", "Headphones", "Keyboard", "Legos", "Media Cases", "Mouse", "Shoe", "Stuffed Toys", "Toys". Objects that do not have one of those categories assigned were not considered. For the "duplicate objects" variant, the exact same object was resampled multiple times. Otherwise, the same settings as for MOVi-C were used.

**PASCAL VOC 2012** We train on the "trainaug" variant with 10 582 images. The trainaug variant is an unofficial split that has been used by prior work (Van Gansbeke et al., 2021; Melas-Kyriazi et al., 2022). It consists of 1 464 images from the segmentation train set, and 9 118 images from the SBD dataset (Hariharan et al., 2011). For object discovery and segmentation, we evaluate on the official segmentation validation set, with 1 449 images. For object localization, we evaluate on the detection "trainval" split (joint official training and validation splits), with 11 540 images, as standard for this task (Simeoni et al., 2021; Melas-Kyriazi et al., 2022). For unsupervised object segmentation, we use the 20 object classes plus one background class. For object discovery and segmentation, unlabeled pixels are ignored during evaluation.

**COCO**    We train on the COCO 2017 dataset with 118 287 images, and evaluate on the validation set with 5 000 images. For object discovery, we use both instance and segmentation masks, converting instance masks into segmentation masks using a per-pixel $\arg\max$ over classes. Overlaps between instances are ignored during metric evaluation, and crowd instance annotations are not used. For unsupervised semantic segmentation, we use the COCO-Stuff annotations (Caesar et al., 2018), which contain the original 80 "things" classes from COCO, plus 91 "stuff" class covering background. In this labeling, overlaps have been manually resolved. We ignore pixels belonging to crowd instances during evaluation. For unsupervised object segmentation, we use the 80 "things" classes, and treat the "stuff" classes as background. For unsupervised scene decomposition, we use the COCO-Stuff 27 labeling (Ji et al., 2019; Cho et al., 2021), resulting from merging the 80 "things" classes into 12, and the 91 "stuff" classes into 15 super-classes, and the COCO-Stuff 171 labeling, using all "things" and "stuff" classes. For object localization, we evaluate on COCO-20k, with 19 817 images. COCO-20k is a subset of COCO 2014 training and validation images, filtered to remove images with only crowd instances and removing crowd annotations.

**Evaluation Settings**    For scene decomposition on COCO-Stuff 27 and COCO-Stuff 171, we take a center crop of the image after resizing the minor axis to 320 pixels and evaluate the masks at $320 \times 320$ resolution to be consistent with prior work (Ji et al., 2019; Hamilton et al., 2022). For object discovery on PASCAL VOC 2012 and COCO, we use the same settings. For object discovery on MOVi, we use the full image (at $128 \times 128$ resolution). Again to be consistent with prior work, for object localization on PASCAL VOC 2012 and COCO-20k as well as object segmentation on PASCAL VOC 2012 and COCO, we evaluate the *full* image after resizing to $224 \times 224$, ignoring aspect ratio. Our model proved to be robust to the aspect ratio distortion to some degree; however, we believe that results can be further improved, e.g. by taking the distortion into account during training, or by using multi-crop evaluation. Predicted soft probability masks are converted to hard masks using the $\arg\max$.

**Metrics**    For *foreground adjusted rand index (FG-ARI)*, we use object/instance masks and only evaluate pixels in the foreground. We treat unlabeled pixels (e.g. on PASCAL) and pixels from overlapping instance masks (e.g. on COCO) as background (i.e. they are not evaluated).

For *mean best overlap (mBO)*, we use the object masks on MOVi, and both instance and segmentation masks on PASCAL/COCO. Each ground truth mask (excluding the background mask) is assigned the predicted mask with the largest overlap in terms of IoU. mBO is computed as the average IoU over the assigned mask pairs. We do not count unlabeled pixels (e.g. on PASCAL) and pixels from overlapping instance masks (e.g. on COCO) when computing IoU.

# G    ADDITIONAL EXAMPLES

We include additional mask predictions of our model for MOVi-C (Fig. 19), for MOVi-E with 24 (Fig. 20) and 11 slots (Fig. 21), for PASCAL VOC 2012 (Fig. 22) and COCO (Fig. 23, Fig. 24, Fig. 25, Fig. 26).

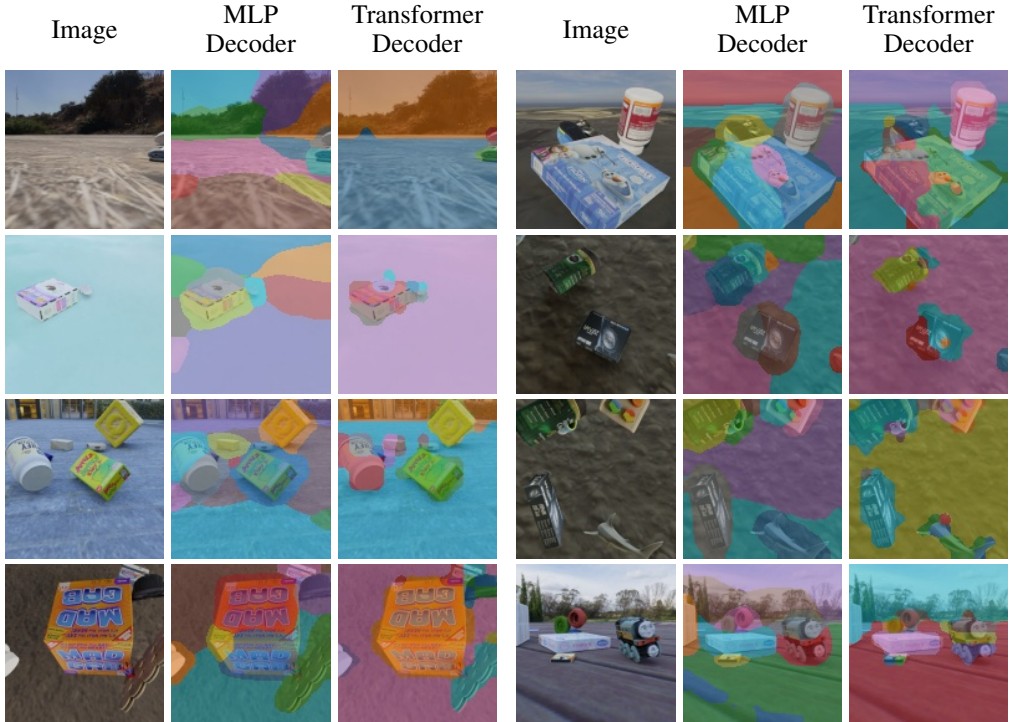

Figure 19: Masks on MOVi-C produced by our method, using 11 slots and a ViT-B/16 encoder. We show predictions from the MLP and the Transformer decoder.

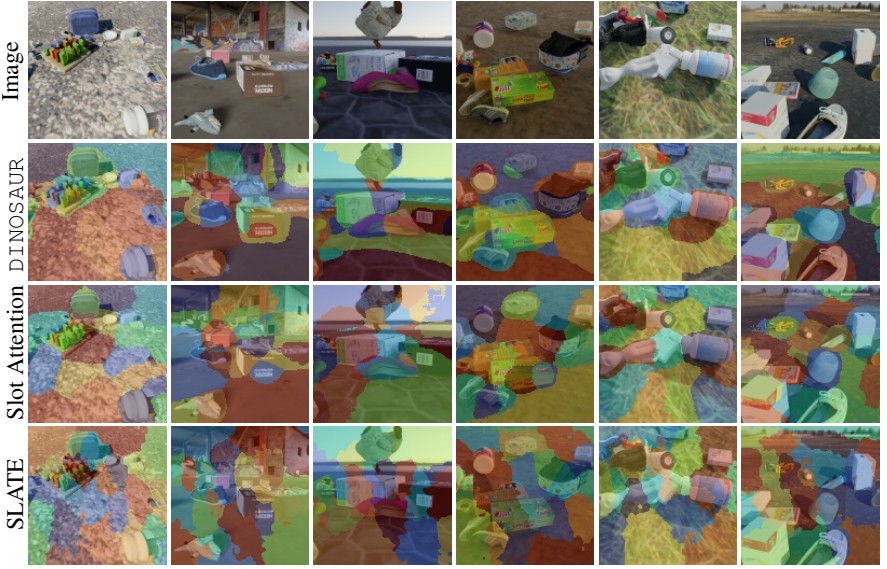

Figure 20: Masks on MOVi-E using 24 slots. Our method uses a ViT-B/8 encoder and the MLP decoder.

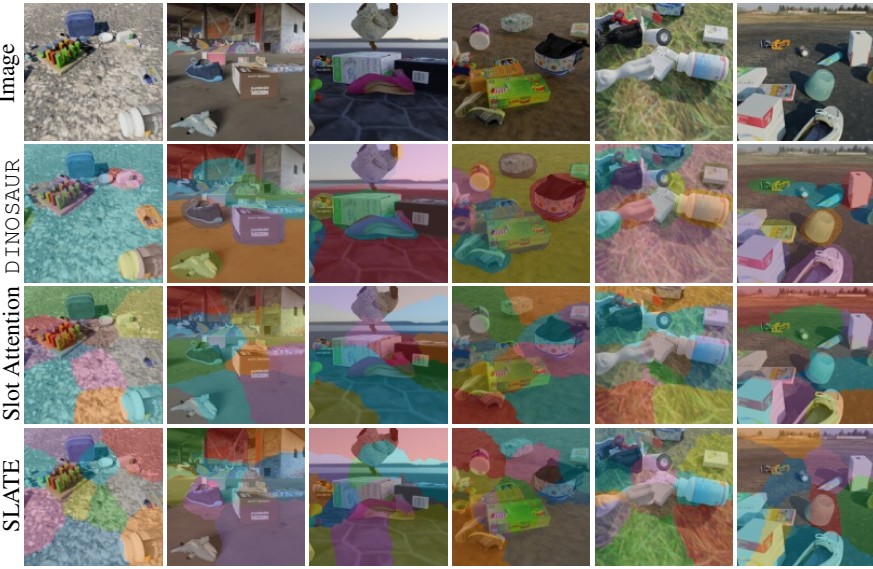

Figure 21: Masks on MOVi-E using 11 instead of 24 slots (quantitative results in Table 13). Our method uses a ViT-B/8 encoder and the MLP decoder.

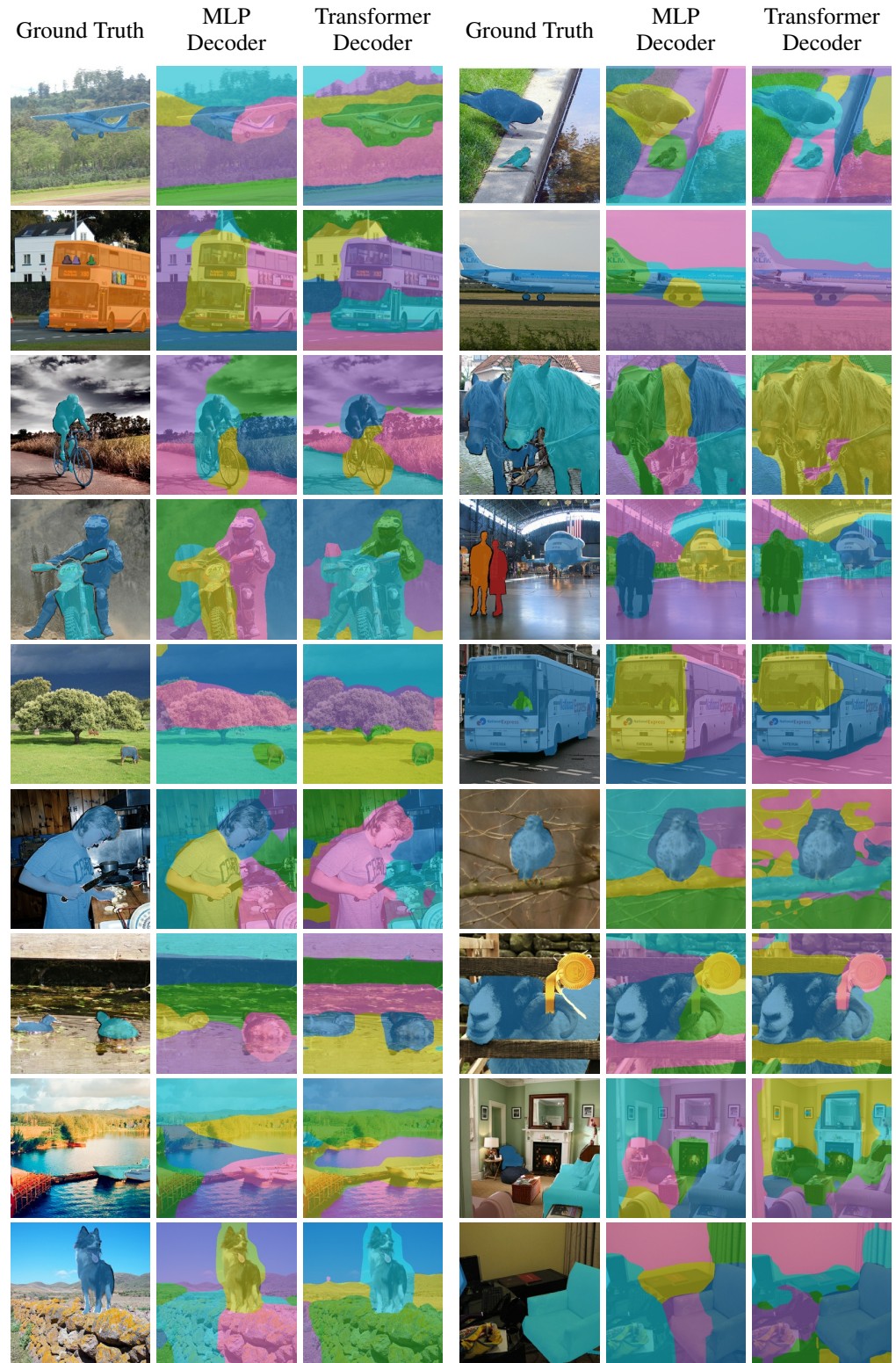

Figure 22: Masks on PASCAL VOC 2012 produced by our method, using 6 slots and a ViT-B/16 encoder. We show predictions from the MLP and the Transformer decoder.

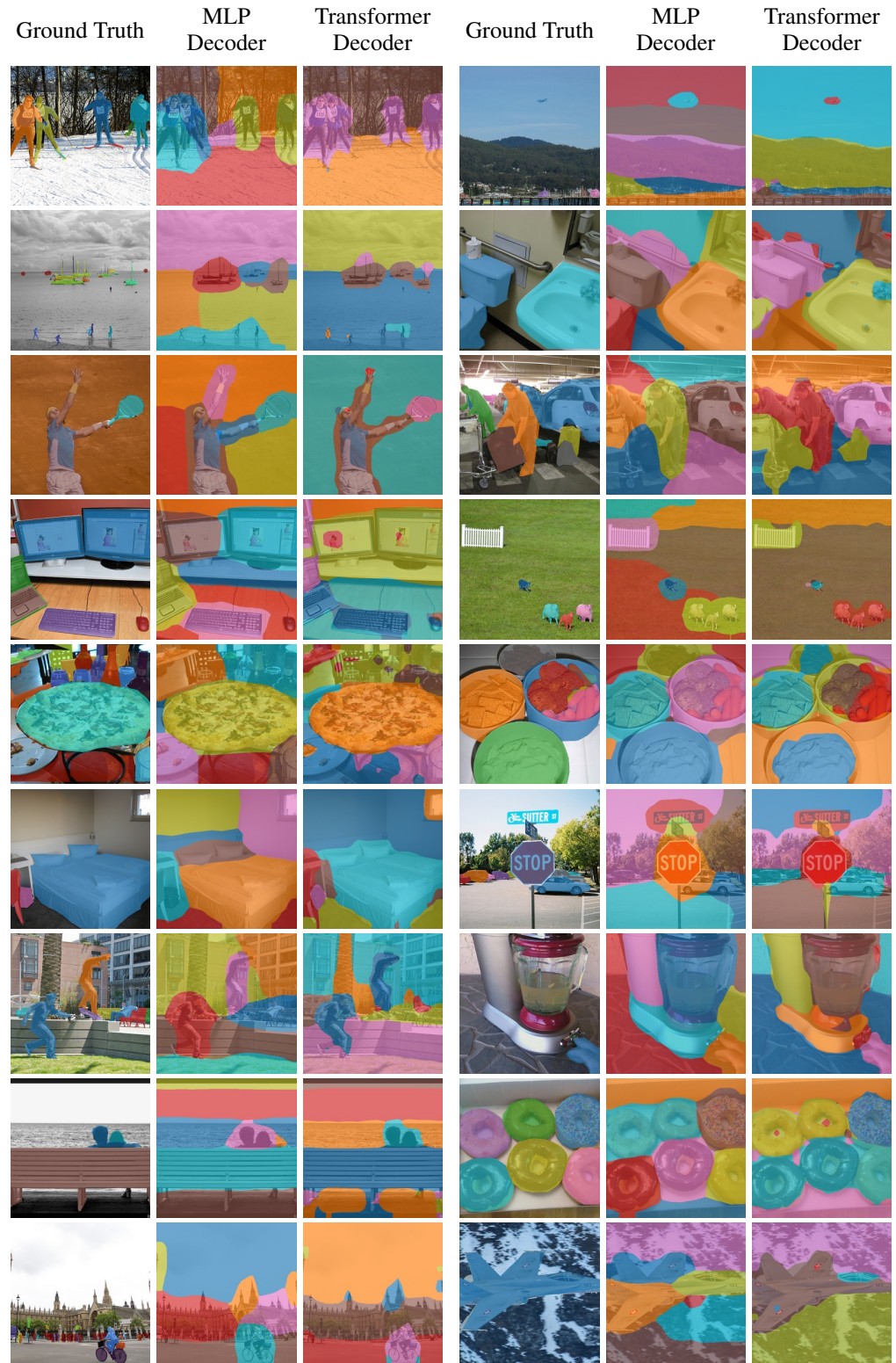

Figure 23: Masks on COCO 2017 produced by our method, using 7 slots and a ViT-B/16 encoder. We show predictions from the MLP and the Transformer decoder.

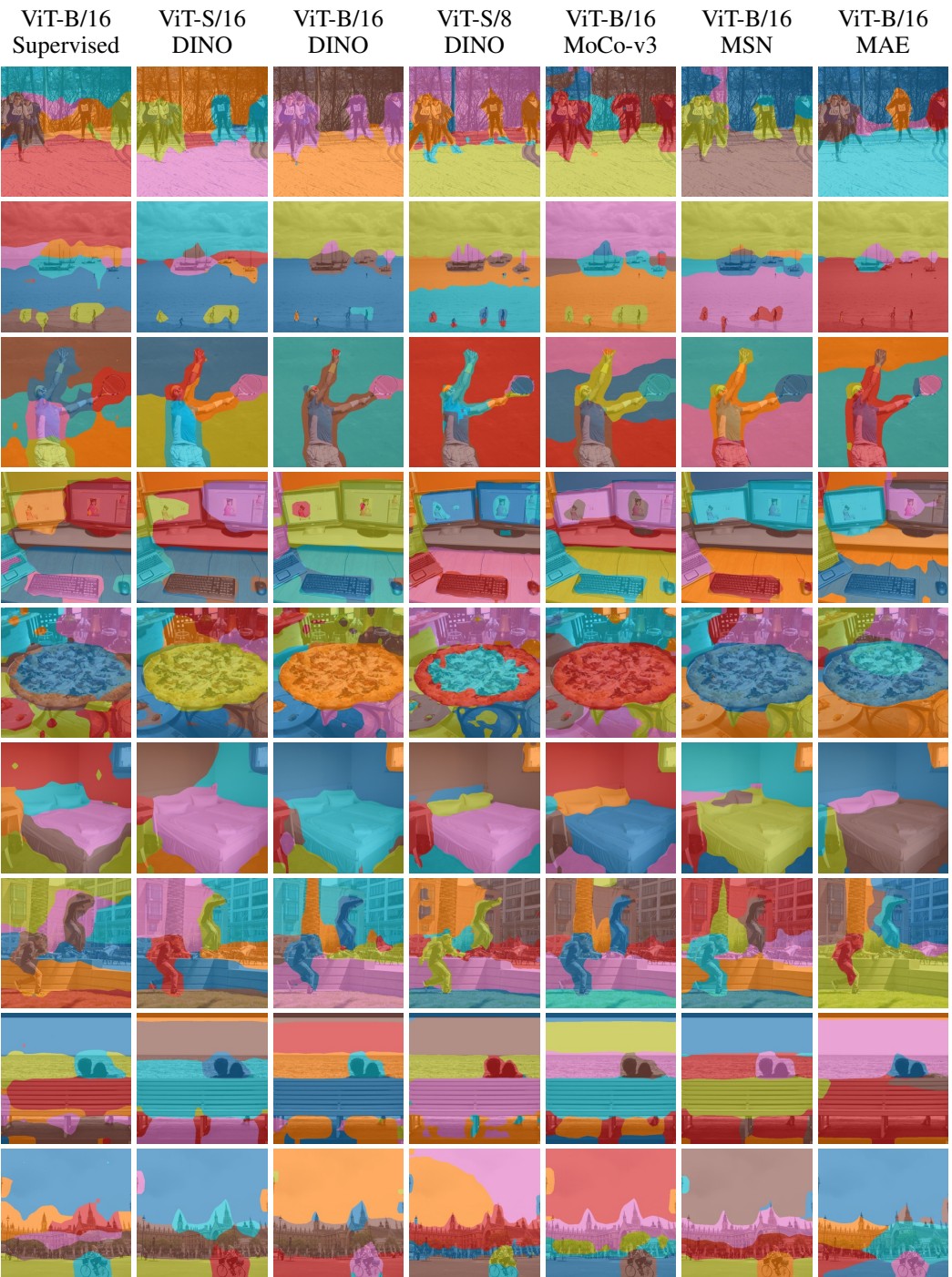

Figure 24: Masks on COCO 2017 produced by our method, using different encoders and pre-training schemes on COCO 2017 (cf. Table 11). All models use 7 slots and the Transformer decoder.

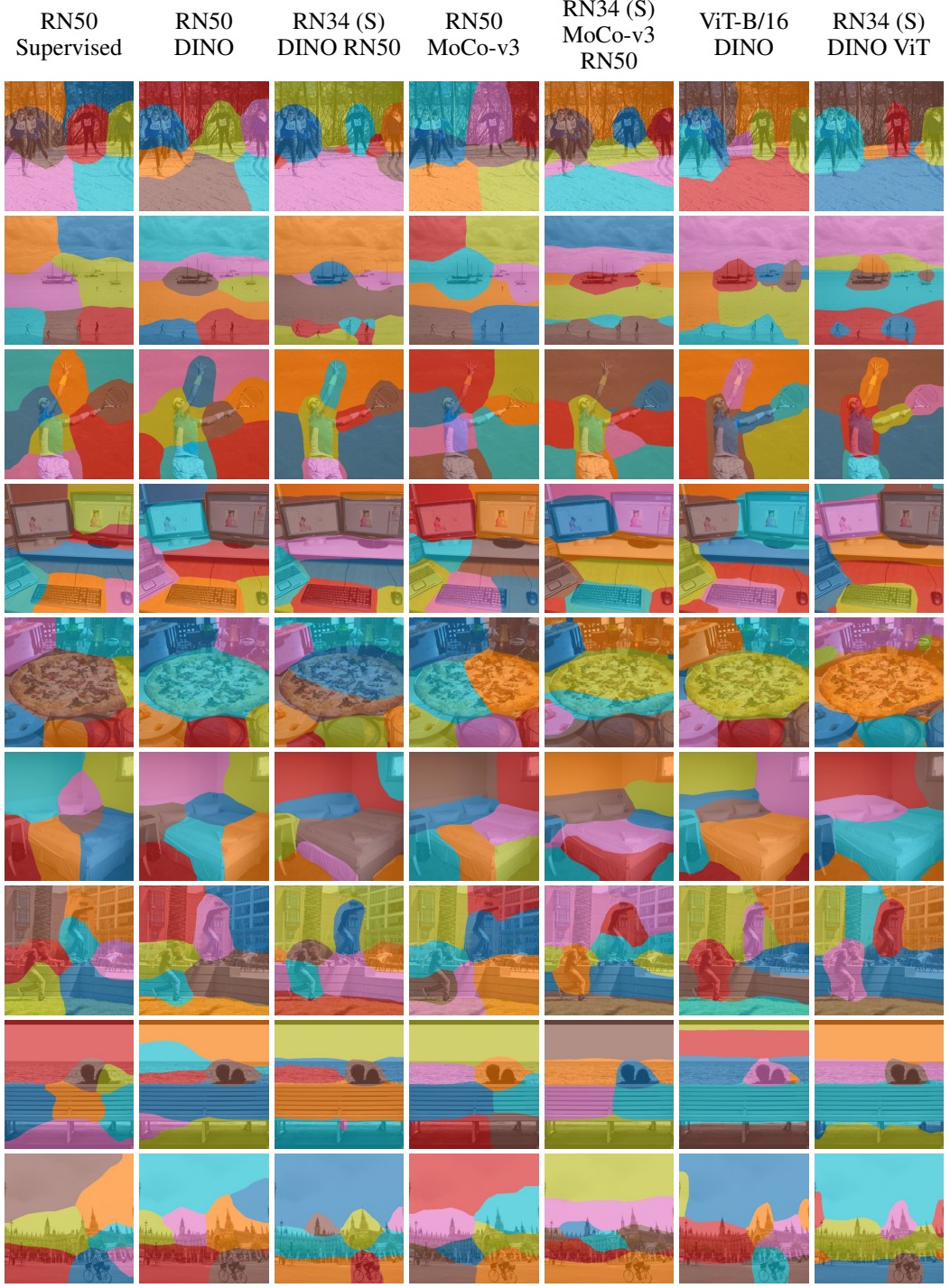

Figure 25: Masks on COCO 2017 produced by our method, using different encoders and pre-training schemes on COCO 2017. ResNet34s (RN34) were trained from scratch ("S"), by reconstruction of ResNet50 (RN50) or ViT-B/16 features (cf. Table 12). All models use 7 slots and the MLP decoder.

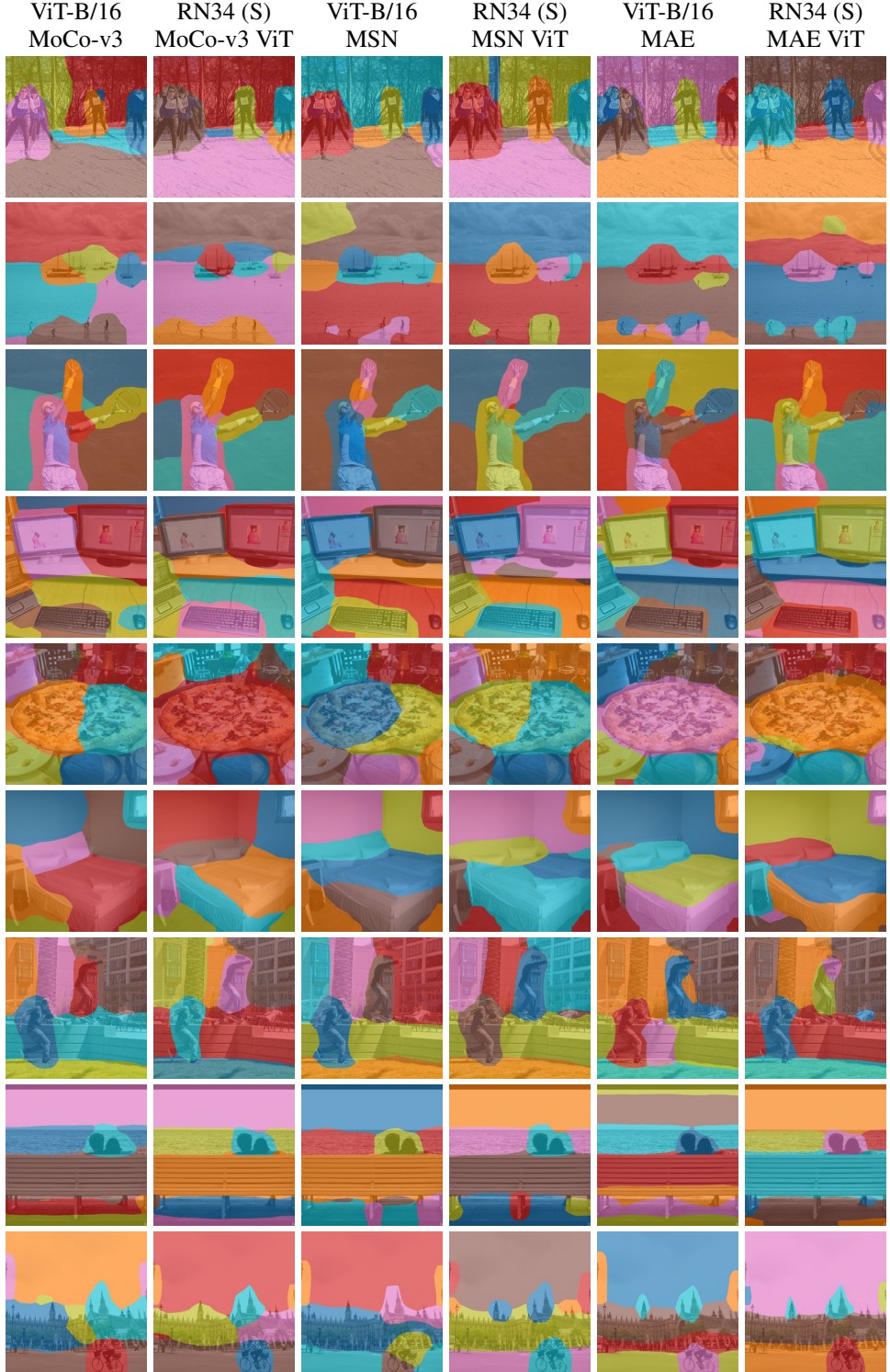

Figure 26: Masks on COCO 2017 produced by our method, using different encoders and pre-training schemes on COCO 2017. ResNet34s (RN34) were trained from scratch ("S"), by reconstruction of ViT-B/16 features (cf. Table 12). All models use 7 slots and the MLP decoder.

