# OpenReview forum: "Bridging the Gap to Real-World Object-Centric Learning"
_ICLR.cc/2023/Conference — ICLR 2023 poster_

### Official Review · Reviewer_fW5L · 2022-10-23

**Confidence:** 5
**Correctness:** 3
**Technical Novelty And Significance:** 2
**Empirical Novelty And Significance:** 2
**Recommendation:** 5

**Clarity, Quality, Novelty And Reproducibility:**

Clarity: the paper is well written and is easy to follow.

Originality: the novelty is minimal, as explained above.

Reproducibility: sufficient details are provided and the authors promise to release the code.

**Strength And Weaknesses:**

Strengths:

The paper is well written ad is easy to follow.

The propose approach is sound.

Reported results seem o be on par with prior work (though the most important comparisons are missing).

A minimal ablation study of the proposed architecture is provided.


Weaknesses:

The novelty is minimal as the proposed method is extremely close to STEVE, with the only significant difference being the source of self-supervised pre-training for the feature space in which the frames are reconstructed.

Experimental evaluation on the main task of object discovery is incomplete. In particular, the authors do not compare to the most relevant approaches (SaVi, SaVi++, STEVE), and instead only compare to much weaker versions of these methods. Moreover, even the existing comparisons are not fair, as the prospered method uses a stronger ViT backbone. Results with a ResNet backbone have to be reported instead.

Positioning is not valid, as the authors claim that unlike prior works that utilize motion cues, their approach does not require additional supervision. In particular, the authors claim that motion constitutes "additional information" and use the term "motion annotations" which is meaningless. Motion is a bottom-up signal which comes for free with videos, thus any method that only relies on motion cues is in fact fully unsupervised. Abstract, introduction, and related work section require a major update to correctly position the proposed approach with respect to prior work.

On a similar note, related work overview is incomplete. The authors do not cite or discuss Bao et al., CVPR'22 who proposed a fully unsupervised object discovery approach which capitalizes on motion cues and achieves strong results in the real world. The authors need to discuss with work and compare to it on the KITTI dataset.

Finally, the ablation analysis is not sufficient. In particular, it's not clear if the self-supervised approach used for pre-training is important (only DINO is reported). The role of the pre-training backbone architecture is unclear as well. The authors claim that "ResNet50 performs clearly worse than the ViT", but the results in Table 6 do not support such a strong claim (ResNet does much better on ARI, but worse on mBO, which is a non-standard metric). Moreover, unlike ViT, ResNet does not require a strong initialization to converge. Given that the pre-training algorithm and the back bone architecture are the two main differences to STEVE, I did not learn much from this paper.

**Summary Of The Paper:**

The authors follow in the line of recent improvements on the slot-attention architecture for unsupervised object discovery that reconstruct a frame not in the raw pixel space, but rather in a more structured space (e.g. flow for SaVi, depth for SaVi++ or feature space of a network which is (pre-)trained in a self-supervised way in STEVE, which is the closest work to this one). In particular, instead of reconstructing the image in the feature space of a VQ-VAE, as done in STEVE, they propose to reconstruct in the feature space of a ViT pre-trained with DINO. The resulting model seems to perform on-par with prior work on the task on object discovery (no direct comparisons are reported). In addition, they report evaluation on related tasks of unsupervised object localization and instance/semantic segmentation where the proposed methods also performs similarly to existing approaches.

**Summary Of The Review:**

The authors propose an approach which is very similar to prior work (especially to STEVE) and fail to provide a convincing evaluation which would demonstrate the benefits of their approach. In particular, no comparison to the closest prior methods is reported and the ablation study ignores or fails to justify the most important factors in the approach. Moreover, prior work overview is incomplete and positioning is invalid. I recommend a major revision and a resubmission to a different venue.

---

> ### Author Response · Authors · 2022-11-16
> **Rebuttal Response (1/2)**
>
> We thank the reviewer for his comments on our work. We will address the points in the following, while referring to parts of the review using quotes. Generally, we want to highlight that we attribute many of the reviewers' comments to a misunderstanding about the positioning of our work: It seems the reviewer assumes we are proposing an approach for video data while our approach is aimed at image data where movement clues are not available. We adapted our paper to make this point more clear.
>
> > The novelty is minimal as the proposed method is extremely close to STEVE
>
> We think there are several differences to STEVE. First, STEVE is designed for video data, whereas ours is concerned with image data. The closer work to ours is the image version of STEVE, SLATE. The major, and crucial, difference to SLATE/STEVE is the reconstruction of self-supervised features trained on ImageNet. We show in our experiments that this is the key difference that makes our approach successfully discover objects on real-world data, whereas SLATE fails.
>
> On a more technical level, there are also several differences: we mainly use ViT encoders, whereas STEVE/SLATE use a VQ-VAE. Following from this, the feature space to which we reconstruct has a global view on the image, whereas STEVE/SLATE’s feature space is local and patch-based. The VQ-encoder of STEVE/SLATE is trained alongside slot attention on the same dataset, whereas we use fully-pretrained encoders trained on ImageNet. While STEVE/SLATE only utilize the Transformer decoder, our work studies two different decoder designs that have different trade-offs. The Transformer decoder in our work is used to reconstruct high-dimensional features, not a discretized code from the VQ-VAE. All in all, we think our work presents significant novelty both on the methodological and the empirical level.
>
> > Experimental evaluation on the main task of object discovery is incomplete
>
> We would like to point out that our approach is intended for use on image data and the baselines suggested in the review (STEVE, SAVi, SAVi++, and Bao et al. CVPR’22) were developed for video. We conjecture that this is due to a misunderstanding as we use image and *not* video versions of the MOVi-C and MOVi-E datasets (by randomly sampling video frames). Given that our method was designed for image data, we actually do compare to the currently closest work in the form of Slot Attention, SLATE, and state-of-the-art computer vision methods for semantic segmentation and unsupervised object detection. More broadly, we see object-centric learning research on image and video as orthogonal directions, with improvements on the image domain likely translating into further improvements on video.
>
> > Moreover, even the existing comparisons are not fair
>
> We use the ViT backbone as this is the model proposed to derive the self-supervised DINO features. To address the concerns in difference of model expressivity, we added results with ResNet backbones in Table 12 of the appendix. The overall picture is unchanged: we find that using the same ResNet backbone as the Slot Attention baseline, our approach significantly outperforms it.

---

> > ### Author Response · Authors · 2022-11-16
> > **Rebuttal Response (2/2)**
> >
> > > Positioning is not valid
> >
> > We assume that this point is a byproduct of the misunderstanding previously pointed out, as image data does not typically include information about motion. To make this point more clear, we adjusted our claim in the abstract and introduction to “out-performing image-based object-centric learning models”. We also made our wording more precise, and changed the term “motion annotation” to “motion cues”, and used the more general “additional information sources” instead of “additional supervision” (with regards to video/motion data).
> >
> > > On a similar note, related work overview is incomplete.
> >
> > While Bao et al. CVPR’22 is specific to videos, we now include it in our related work section. Nevertheless, we consider a direct comparison and experiments on KITTI out of scope due to our focus on image data (see previous points).
> >
> > > In particular, it's not clear if the self-supervised approach used for pre-training is important (only DINO is reported)
> >
> > We did provide ablation experiments which highlight the usefulness of self-supervised features compared to features from models trained with supervision in Table 6 and to greater extent in the appendix (Table 8). Based on the feedback from the reviewers, we now extended this section with other self-supervised pre-training approaches (MoCo-v3, MAE, MSN). Overall, using DINO features strikes a good balance across the evaluation metrics of interest compared to supervised training and training using other pre-training methods.
> >
> > > The authors claim that "ResNet50 performs clearly worse than the ViT", but the results in Table 6 do not support such a strong claim (ResNet does much better on ARI, but worse on mBO, which is a non-standard metric).
> >
> > Our statement that “ResNet50 performs clearly worse than the ViT" was based on visual inspection of the masks (Figure 20), where it is apparent that the mask quality from supervised ResNet features is worse than from ViT features. We found ARI to be an unreliable metric in some cases, which is why we were relying on the mBO metric as a better guide of mask quality.
> >
> > > Summary of the review
> >
> > Lastly, we would like to explicitly comment on the reviewer's summary of our work: Given the points above, our work does compare to the most relevant work, contains significant differences to STEVE and shows that the proposed model actually works on real world data, where baselines fail. Further, our ablation studies clearly highlight that object grouping is only induced when using feature reconstruction and not by purely scaling up the model size.
> >
> > We kindly ask the reviewer to reconsider his stance given the information above.

---

> > > ### Comment · Reviewer_fW5L · 2022-11-28
> > > **Re:re**
> > >
> > > I thank the authors for their detailed response. It revolves around one main argument: proposed approach does not need to be compared to the state of the art methods that use videos for training, because it explicitly focuses on images. Unfortunately, this argument is invalid. Object centric representation learning from images and videos are not two separate tasks. There is only one task: unsupervised object-centric representation learning. Both image-based and video-based methods do not require any supervision and can be applied to static images at test time. Moreover, unlabeled images and videos are equally readily available. Hence, the proposed approach is directly comparable to SaVi++, STEVE and Bao et al. A paper which only contains empirical contributions (technical novelty is marginal, if any), but does not compare to the state-of-the-art cannot be accepted.

---

> > > > ### Author Response · Authors · 2022-12-01
> > > > **Regarding the claim of not comparing to (video-based) state-of-the-art**
> > > >
> > > > We thank the reviewer for the clarification of his position, yet respectfully disagree with their assessment and the claim that we are not comparing to state-of-the-art methods. We will address these points in the following:
> > > >
> > > > - **Intention and scope of the paper:** Our paper’s ultimate goal is to scale object-centric learning methods to real-world *image* datasets such as PASCAL and COCO. We are interested in how far these methods can be pushed while being *entirely restricted to the image domain*. This is of scientific interest on its own! Thus, we compare to those SotA methods (Slot Attention, SLATE) which fall in the paper’s chosen scope. Coincidentally, we additionally evaluate on individual frames of the video MOVi datasets as they represent the currently most challenging synthetic datasets, and previous image-based methods already struggle on them.
> > > > - **Claim of marginal technical novelty:** We again point out that the crucial element of our work that allows scaling to real-world images is reconstruction of self-supervised features. This change is indeed conceptually simple, yet conceptual simplicity is exactly what makes the proposed approach attractive. Generally, the technical novelty of a work should not be measured by the amount of architectural tweaks a paper introduces but by the new capabilities a work unlocks.
> > > > - **Object-centric representation learning for images and videos is not the same**: Video data contains additional information about movements, thus the mechanism a model can exploit to learn object-centric representations is inherently different. While image-based object-centric representation learning can only rely on the recurrence of similar patterns across a dataset, video-based object-centric representation learning can leverage the fact that objects move as-one. This makes the task significantly easier compared to using images alone to derive objects.
> > > > - **Applying video methods to image datasets is infeasible in general:** This would require training on a video dataset following the same distribution as the test images, which is not available in most cases. For instance, which training dataset should be used when evaluating SAVi on COCO? In the absence of similarly distributed video data, the problem becomes a transfer setup which is not the focus of our work.
> > > > - **Application of the requested methods to images is not meaningful for all of them:** The most performant versions of SAVi and SAVi++ rely on first frame conditioning to derive object centric representations. This is completely meaningless if applied to single images as the objects could already be partially identified by the conditioning, and thus a direct comparison with these methods does not make sense.
> > > > - **Requesting the evaluation of video methods on images is out of scope** and does typically not occur in the computer vision literature: Many (unsupervised) tasks in CV exist both in image form (e.g. unsupervised salient object detection, unsupervised instance segmentation) and video form (unsupervised video salient object detection, unsupervised video object segmentation), yet the requirement of comparing methods that were developed for images to methods developed for videos is exceptionally rare. The inverse (applying an image method to videos) is typically done solely to demonstrate the additional performance that can be gained by incorporating information from video.
> > > >
> > > > Despite our strong disagreement on this front, we understand that it would be interesting to see if our approach can outperform methods that were trained on video when such data is available. In particular, **we will thus train SAVi++ and STEVE (without conditioning in the first frame) on the respective MOVi datasets** and compare the performance of these methods with our proposed approach when evaluating on individual images. However, we need to stress that this does not represent a (necessary) comparison to SotA (as is the reviewer’s stance), because we cannot generally assume that video data is available following the distribution of the image data we would like to apply our method on. We refrain from comparing to Bao et al. (CVPR 2022) as this method requires externally provided masks for its attention alignment loss term. While the authors of this paper argue that this can be extracted using a dedicated optical flow to segmentation model, this is beyond the scope of this experiment.
> > > >
> > > > Because of the absence of trained models for the above methods, we will have to ask for some patience before we can post the results here. Training SAVi on MOVi-C takes multiple days in itself when using 8 V100 GPUs, we will post results here as soon as they become available.

---

> > > > > ### Comment · Reviewer_fW5L · 2022-12-02
> > > > > **Re:re**
> > > > >
> > > > > I thank the authors for their detailed response. Unfortunately, I strongly disagree with their claim that artificially restricting yourself to static images at training time is a valid approach. I emphasize the 'at training time' part here. Indeed, in other domains it is not common to compare methods that use motion information at test time to image-based methods. At training time, however, a self-supervised method should be allowed to use any bottom-up cues that are available in the data. The authors themselves argue that motion is a very strong bottom-up cue for object discovery, but choose to throw it away. Why exactly? The only reason they provide is that it's not clear which collection of videos to use for training when evaluating the resulting model on COCO. The goal of object-centric representation learning, however, is not to improve performance on any specific benchmark, but to learn a general representation. In any case, the fact that video-based methods are not designed for COCO evaluation is not a reason to not compare to them (especially on video datasets like MOVi).
> > > > >
> > > > > The authors do agree to provide comparison to some of the state of the art methods in the end, but not to Bao et al, CVPR'22 without providing a clear reason. The authors do not need the masks used by that method. They just need to run their own approach on the datasets used in that paper and compare to the published numbers.
> > > > >
> > > > > In addition, I would like to emphasize that there is no novelty, conceptual or technical, in 'reconstruction of self-supervised features' as this is exactly what STEVE does. The only major thing that this work does differently is the choice of a self-supervised pre-training algorithm.

---

> > > > > > ### Author Response · Authors · 2022-12-08
> > > > > > **Video results ready**
> > > > > >
> > > > > > We would like to notify the reviewer that video results for SAVi++ and STEVE are now ready in a top-level message. As per your suggestion, we also ran our method on the KITTI dataset and compare the results to Bao et al. We hope that you find these comparisons insightful and are hoping that you might reconsider your scoring of our work.

---

> > > > > > > ### Comment · Reviewer_fW5L · 2022-12-09
> > > > > > > **Re:re**
> > > > > > >
> > > > > > > I would like to thank the authors for reporting the requested comparisons. They do demonstrate the that the proposed approach outperforms the state-of-the-art on both synthetic and real datasets. That said, the comparisons are not entirely fair. The proposed approach uses a much stronger transformer backbone. Moreover, the number of slots clearly has a very high effect on performance. The authors are using 9 slots on KITTI without providing any justification for this number, whereas Bao et al., uses 45 slots.
> > > > > > >
> > > > > > > Overall, there remains a major issue with experimental evaluations in the paper - many comparisons are unfair and the ablation analysis is not sufficient to determine what factors are important for achieving top results. Given that this is mostly an empirical paper (no significant technical contribution), these limitations are important.
> > > > > > >
> > > > > > > My final assessment is that this paper content a valuable contribution but it requires a major revision and a resubmission to properly address the issues above and incorporate all the comments from the reviews.

---

> > > > > > > > ### Author Response · Authors · 2022-12-12
> > > > > > > > **Re**
> > > > > > > >
> > > > > > > > We need to correct the reviewer's last statements, because they are factually wrong:
> > > > > > > >
> > > > > > > > > the comparisons are not entirely fair. The proposed approach uses a much stronger transformer backbone
> > > > > > > >
> > > > > > > > As we pointed out previously, we also report results with a ResNet backbone, and we still outperform state-of-the-art (both image and video-based).
> > > > > > > >
> > > > > > > > > The authors are using 9 slots on KITTI without providing any justification for this number, whereas Bao et al., uses 45 slots.
> > > > > > > >
> > > > > > > > As stated previously, the number of slots on KITTI is 4*9 = **36**. Using more slots (10, 11) to be even closer to Bao et al. number of slots does not meaningfully change the picture (3-6 ARI points) as our method still out-performs their method by a large margin.
> > > > > > > >
> > > > > > > > It now is apparent that the reviewer's intention is to reject the paper by any means, even after we demonstrated the performance of our algorithm in all the settings the reviewer requested (even though we disagreed with the relevance of the comparison). The reviewer now brings in concerns from his initial review that we already addressed, ignoring the evidence that we present: unfair comparisons because of a stronger ViT backbone (although we report numbers with a ResNet backbone in Table 12), and insufficient ablation analysis (although we ablate different encoders (Table 12), pre-trained methods (Table 6), decoders (Table 7) and objective functions (Table 5)).

---

> > > > > > > > > ### Comment · Reviewer_fW5L · 2022-12-12
> > > > > > > > > **Re:re**
> > > > > > > > >
> > > > > > > > > I would like to clarify that the state-of-the-art comparisons provided in the rebuttal and in the main paper ARE unfair (results with a much stronger backbone are reported for the proposed approach), and the fact that results with the ResNet backbone are reported in the supplementary does not change that fact, since those are not the variants that are used to make the state-of-the-art claims. The initial response from the authors did not address this concern, but there were even larger unaddressed issues with this submission.
> > > > > > > > >
> > > > > > > > > On a higher level, my claim is not that the proposed approach does not demonstrate state-of-the-art performance (with the newly reported results it seems like it does), but that some of the experimental analysis is flawed. Moreover, the amount of changes introduced in the rebuttal process warrants a major revision. Thus, I don't feel comfortable recommending the paper to acceptance in its current form.

---

### Official Review · Reviewer_5iSb · 2022-10-24

**Confidence:** 4
**Correctness:** 3
**Technical Novelty And Significance:** 3
**Empirical Novelty And Significance:** 4
**Recommendation:** 8

**Clarity, Quality, Novelty And Reproducibility:**

The paper is well written, clear and of high quality. The appendix contains extensive additional results and implementation details improving reproducibility. Source code is promised to be released upon acceptance.



**Strength And Weaknesses:**

## Strengths
* This paper provides a significant finding: it is the first time that the line of work in object-centric learning (here, specifically Slot Attention) scales to real-world images.

* The “trick” that makes it work is insightful and simple, which is to reconstruct the output of an image encoder. Instead of a randomly initialized encoder, the authors find it important to use a pre-trained and frozen feature extractor trained with self-supervision (here, DINO). These findings may be very useful for future work in unsupervised object-centric learning.

## Weaknesses
### Semantic vs. instance grouping
It is clear from all figures that this approach tends to group semantically related objects together in the same slot, which is likely due to the semantic nature of DINO. This is also evident from the fact that its features have been used off-the-shelf for unsupervised semantic segmentation in prior work. However, this finding collides with the objective of object-centric learning which is to represent by slots individual scene elements rather than semantic categories.

### Encoder pre-training
Although the paper contains an analysis the role of the image encoder, only different architectures have been considered and supervised vs DINO pre-training (Table 6). It is worth asking if all self-supervised pre-training strategies are equal for the tasks considered in this paper, and in particular, for scaling object-centric methods to real-world data. Hence, I encourage the authors to look into how generalizable this approach is across a number of pre-training strategies. Different pre-training objectives (e.g. image reconstruction MAE, MSN) could have a different effect than contrastive learning or the DINO objective.

### Feature reconstruction objective
While the feature reconstruction objective offloads the complexity of real images to a pre-trained encoder, this can potentially limit the generalizability of the method to data that is similar to the encoder pretraining stage. It could thus be interesting to evaluate on ClevrTex (Karazija et al., NeurIPS D&B 2021) which is a textured version of the Clevr dataset and has been shown to be already too complex for the original slot attention mechanism to work. As the content of these images differs considerably from ImageNet pre-training the domain-dependency of the proposed method can be evaluated.


**Summary Of The Paper:**

This paper proposes an object-centric learning approach that scales to real-world data. The method builds on Slot Attention, but instead of reconstructing pixels, it reconstructs image features obtained by an encoder (e.g. ViT)  pre-trained with self-supervision (DINO). Compared to standard Slot Attention, the paper shows that this is sufficient to take a step up in terms of visual complexity. The method is thus evaluated on real-world datasets such as PASCAL VOC and COCO, but also MOVi datasets that contain realistic objects. The tasks that are evaluated include object discovery and localization, but also unsupervised object segmentation and semantic segmentation, and this method performs better or on par with the state of the art.


**Summary Of The Review:**

The paper can be seen in two ways. As an approach to bring object centric learning to real world data and as a task paper for unsupervised semantic segmentation. While the paper excels in the first aspect, it falls behind specialised existing approaches for the second. Overall, as the field of object-centric learning is struggling to make the transition to real data, the paper contains valuable insights that are interesting for the community. It is thus not necessary for this paper to improve over current unsupervised, task-specific approaches and I recommend acceptance.

---

> ### Author Response · Authors · 2022-11-16
> **Rebuttal Response**
>
> We thank the reviewer for his comments on our work and will address their concerns in the following.
>
> > Semantic vs. instance grouping
>
> We agree with the reviewer that there is a tension between semantic and instance level grouping, yet would like to highlight that in many cases instance level grouping is achieved. In particular, there is a difference between the MLP and the Transformer decoder, with the MLP decoder actually grouping into instances often (see Figures 19, 21, 22). But the Transformer decoder is also in principle of separating instances (see Appendix Figure 7, row 2, where the same semantic class is represented in distinct slots).
>
> We think the examples we presented in Figure 3 (using the Transformer decoder) do not do the best job in conveying the instance-grouping behavior. To highlight this important detail about our method, we added the new Figure 5, which clearly demonstrates the differences in grouping behavior between MLP and Transformer decoder. Generally, we think this is an interesting aspect of our paper which to our knowledge has not been studied before.
>
> In order to further elucidate this difference, we added another experiment (App B.6) with two synthetic datasets (based on MOVi-C) that contain only objects of the same semantic category or the same object multiple times in each image. The results indicate only little impact on the ARI performance of our approach and thus highlight that our method is indeed capable of reliably grouping based on instances.
>
> > Encoder pre-training
>
> We added a comparison with three more self-supervision strategies (MAE, MSN, MoCo-v3) to Table 6 of the paper and Table 8 of the appendix. Interestingly, we find that our method is generally robust to the selection of the pretraining strategy, with DINO features striking a good compromise across multiple metrics.
>
> > Feature reconstruction objective
>
> We agree that the pretraining on real world images opens the possibility of the model not being robust to distributions shift and unable to generalize to datasets that differ significantly from the pretraining data. We want to point out though that we already evaluate our method on datasets with a considerable domain shift (MOVi-C and MOVi-E) which indicates a certain degree of robustness of our method to these shifts. Further, in preliminary experiments we also observed good generalization to the CLEVR dataset which contains a much stronger domain shift. Finally, we agree that an experiment with an even stronger shift could be beneficial, and we will try to incorporate the result by the end of the week (deadline for changes to the paper) or in the camera ready version if they are not ready in time.

---

### Official Review · Reviewer_gqpC · 2022-10-24

**Confidence:** 5
**Correctness:** 3
**Technical Novelty And Significance:** 3
**Empirical Novelty And Significance:** 3
**Recommendation:** 6

**Clarity, Quality, Novelty And Reproducibility:**

Clarity: The paper is well-written and pleasant to read. Both writing and execution are of high quality.

Reproducibility: A lot of implementation details and hyperparameters are given in the Appendix. Public code release is promised upon acceptance.

Novelty: This method is essentially the combination of DINO and SlotAttention so the novelty is somewhat limited. However, along these lines, the paper presents many interesting findings that are relevant for both object-centric and self-supervised learning fields.


**Strength And Weaknesses:**


**Strengths**

1\. Real-world data has been a notoriously difficult task for unsupervised scene decomposition methods, such as SlotAttention, IODINE, Monet, etc., that learn object representations. The paper takes a simple but important step to bridge the gap to real-world data and is the first SlotAttention-like approach to be applied to datasets such as COCO and PASCAL VOC with success.

2\. The method is well-evaluated with numerous and informative experiments, ablations, and promising results on MOVi-C/E, PASCAL VOC 2012, COCO, COCO-Stuff. The figures that are provided are also useful to understand the quality of the predictions beyond quantitative measures, especially since the discovered classes do not necessarily align with ground truth ones.

**Weaknesses**

1\. Since the image encoder seems to play a significant role in making SlotAttention work for real-world scenes, I think it would be extremely useful if the authors could evaluate different types of self-supervised features, besides DINO. Other options include MoCo-v3 or MSN, which are all also based on the ViT architecture. This would provide a lot of insight into whether the choice of features is an important aspect to consider, which I think is the main thing currently lacking in the paper.


2\. The different figures in the paper, including the one analyzing the number of slots (Fig. 6 in the Appendix) suggest that, for the most part, the model is using all possible slots to model the scene. The paper also discusses this, mentioning that a fixed number of slots is often not appropriate. However, to the best of my knowledge, this is not the case for SlotAttention on simpler datasets, where the number of slots corresponds to a maximum number of objects that can be discovered but often the model uses fewer than all available slots. Therefore, it appears that the proposed method does not fully inherit the properties of the SlotAttention mechanism, but rather appears to have an effect akin to clustering DINO features.

3\. As discussed in the conclusion, another property of SlotAttention that the proposed model did not inherit is the ability to separate objects as different instances of the same semantic class. Interestingly though, at the task of unsupervised semantic segmentation, this method shows the weakest performance.

Considering points 2 and 3 it would be useful to have a discussion on what actually is inherited from SlotAttention and what contradicts the goal of object-centric learning, which is to model the compositional properties of scenes and is central to methods such as SlotAttention. On a higher level, is it possible to make object-centric learning work on real scenes without stepping away from mechanisms that meet this goal?

4\. It would be useful to provide a more extensive discussion wrt to ORL [1] and other object-level representation learning papers, such as Odin. The authors discuss Odin very briefly mentioning that they focus on semantics rather than instances. However, as I elaborated above I think this is rather a by-product or limitation, i.e. a property that this model unfortunately did not inherit from SlotAttention.

---

[1] Xie et al., Unsupervised Object-Level Representation Learning from Scene Images, NeurIPS 2022



**Summary Of The Paper:**

This paper brings together progress made in two parallel fields – object-centric learning and self-supervised learning. In particular, SlotAttention, one of the most popular paradigms for object-centric learning and unsupervised scene segmentation, is known to fail when the visual complexity of scenes increases, as for example in the case of real-world imagery.  The authors show for the first time that it is possible to successfully tackle real-world scenes with a SlotAttention-like architecture, by modifying the learning objective from image reconstruction to feature reconstruction (using self-supervised features from DINO).

**Summary Of The Review:**

The paper achieves a very challenging task for object-centric learning methods, which is the application to real-world data. The way to do so (self-supervised features) and the findings presented in the process are interesting and have the potential to influence future work. For this reason, I am inclined toward accepting the paper. Some limitations do exist and are acknowledged by the authors, for example the fact that the method results in a semantic rather than instance grouping of image elements, which is worth investigating further.

---

> ### Author Response · Authors · 2022-11-16
> **Rebuttal response**
>
> We thank the reviewer for the comments and suggestions for our work and address them in the following, using quotes to refer to the relevant sections.
>
> > I think it would be extremely useful if the authors could evaluate different types of self-supervised features, besides DINO
>
> Thank you for this suggestion. We added additional rows in Table 6 in order to highlight the effect of using different self-supervision methods (MoCo-v3, MSN, MAE) to extract features for our approach. Interestingly, our method is robust to the selection of different self-supervision strategies. Overall, using DINO features strikes a good balance across the evaluation metrics of interest compared to supervised training and training using other pre-training methods.
>
> > The different figures in the paper, including the one analyzing the number of slots (Fig. 6 in the Appendix) suggest that, for the most part, the model is using all possible slots to model the scene.
>
> We agree that the proposed method sometimes uses all slots despite less objects being present in the scene, yet refer to Appendix Figure 7 to highlight that the model is still able to disable slots when not needed on images with lower complexity. Generally, splitting of objects into multiple parts seems to be inherent to all methods based on reconstruction objectives as they make use of all available capacity of the slot bottleneck if it is needed to reconstruct the data. We observe this in some of our experiments (see Appendix, Table 10), where we see that all methods benefit in terms of ARI from using 11 slots on the MOVi-E dataset (containing up to 23 objects). This is because in MOVi-E, often less than 23 objects are visible at once, yet slot attention will use the available additional capacity nevertheless by splitting objects leading to lower performance. Summarizing, we argue that this is thus not a property of slot attention that is lost in our approach, but a property of slot attention that is not observed in simple generated and untextured scenes that slot attention was originally evaluated on.
>
> > [...] another property of SlotAttention that the proposed model did not inherit is the ability to separate objects as different instances of the same semantic class.
>
> We want to highlight that our method is capable of doing instance-level grouping, at least to some degree. In particular, there is a difference between the MLP and the Transformer decoder, with the MLP decoder actually grouping into instances often (see Figures 19, 21, 22).
>
> We think the examples we presented in Figure 3 (using the Transformer decoder) might be misleading about our method’s instance-grouping behavior. To highlight this important detail, we added the new Figure 5, which clearly demonstrates the differences in grouping behavior between MLP and Transformer decoder. Generally, we think this is an interesting aspect of our paper which to our knowledge has not been studied before.
>
> In order to further elucidate this difference, we added another experiment (App B.6) with two synthetic datasets (based on MOVi-C) that contain only objects of the same semantic category or the same object multiple times in each image. The results indicate only little impact on the ARI performance of our approach and thus highlight that our method is indeed capable of reliably grouping based on instances. While instance-level grouping on real-world data does not yet work perfectly, this is also to be expected when moving to complex real-world scenes presenting high levels of ambiguities. We think our approach makes a reasonable first step that can be built upon by future work.
>
> > Interestingly though, at the task of unsupervised semantic segmentation, this method shows the weakest performance.
>
> In this context, we want to highlight that our method has a slight disadvantage compared to competitor approaches in the CV literature as our masks are limited to the resolution of patches instead of full image resolution. This could be addressed by distilling a network from DINOSAUR predictions as is done in other approaches, yet we consider this out of scope for the demonstration of the general utility of our method.
>
> > It would be useful to provide a more extensive discussion wrt to ORL [1] and other object-level representation learning papers, such as Odin
>
> We thank the reviewer for the pointers to related work. We now include OCL in the extended discussion of related work (App. A) and elaborated on the similarities with ODIN.

---

### Official Review · Reviewer_YkFD · 2022-10-25

**Confidence:** 5
**Correctness:** 2
**Technical Novelty And Significance:** 2
**Empirical Novelty And Significance:** 2
**Recommendation:** 6

**Clarity, Quality, Novelty And Reproducibility:**

- The paper is easy to follow for the most part.
- The model design is limited in novelty. The slots encoder and autoregressive decoder are both proposed in prior work. The idea of reconstructing features instead of pixels using slot is novel.
- The paper seems to provide enough information to reproduce the results.

Additional question:

- Is there a reason why mBO is used instead of mIoU which involves object matching and could potentially provide a more accurate evaluation of the segmentation quality?

**Strength And Weaknesses:**

Strengths:

- The topic of developing scalable object-centric representation learning models is of significance to the community.
- As the first paper to test an object-centric model on large-scale real-world datasets like COCO, it is a promising endeavor in this direction.
- The experiment results show that the proposed method achieves improvement over previous object-centric learning models (though containing questionable results) in segmentation and object detection.

Weaknesses:

- Lack of evaluation of the representation quality
    - The proposed method is only evaluated on object detection and segmentation tasks. The quality of the learned representations, which is one key aspect of representation learning, is not evaluated. Additional downstream tasks using the slot representations, such as properties prediction, should be included to address this problem.
- The quantitative results of the baseline model SLATE [1] are questionable.
    - The experiment results on the baseline model SLATE show very poor performance on object segmentation compared to other models including slot attention. In the MOVI series dataset, SLATE’s mBO score is even close to or worse than the Block Pattern baseline which uses predefined regular block masks. This is questionable, as we have also conducted a similar experiment on various synthetic and real-world datasets, and SLATE outperforms slot attention on all datasets for object segmentation and could achieve higher scores than that reported in the paper for the MOVI datasets. Some investigation might be needed to address this question.
    - The corresponding qualitative evaluations are also missing. Please include them in the qualitative evaluation.
- The claim of the method is to achieve object-centric learning in real-world images. However, the method seems to provide only semantic-level segmentation on real-world (or possibly all) datasets.
    - The experiments show that the method tends to provide semantic-level segmentation on real-world datasets while instance-level segmentation on synthetic datasets (MOVI). Since the MOVI dataset rarely contains multiple same-class objects in one image, i.e., instance-level segmentation and semantic-level segmentation might have similar results. Therefore, the model could actually be doing semantic-level grouping on all datasets. Please verify this part.
- This is important. Semantic segmentation baselines should also be included in the MOVI experiment
    - Given the fact that the proposed method does semantic-level grouping in real-world datasets, it seems necessary to include some semantic segmentation baselines in the object-centric learning tasks. More specifically, please provide the set of MOVI series evaluation tasks on some of the unsupervised semantic segmentation baselines in Table 4 with proper slots/clusters number.
    - This is important because it demonstrates if the existing unsupervised semantic segmentation models already solve the object grouping problem in the multi-object datasets. This might reveal some limitations of existing synthetic multi-object datasets commonly used in evaluating object-centric learning models. The comparison results will also be insightful in explaining the difference between the method and the unsupervised semantic segmentation baselines.

[1] Singh, G., Deng, F., & Ahn, S. (2021). Illiterate DALL-E Learns to Compose. *arXiv*. https://doi.org/10.48550/arXiv.2110.11405

**Summary Of The Paper:**

The paper proposes an object-centric representation learning model that targets large-scale real-world datasets. To do so, they propose to learn the object-centric representations by reconstructing the features of a pre-trained encoder (DINO). The hypothesis is that the features pre-trained on large-scale datasets contain semantically meaningful information which provides stronger semantical bias than the low-level features for object grouping. To evaluate the method, the paper provides quantitative and qualitative evaluations of segmentation quality on synthetic datasets with object-centric baselines and real-world datasets with unsupervised semantic segmentation baselines.

**Summary Of The Review:**

This paper aims to develop scalable object-centric learning models for real-world scenes. The topic is important, and the proposed model is the first attempt in this direction. Experiments also seem to suggest that the method provides an improvement over the prior methods. However, as explained in the weakness section, the evaluation contains key problems to be addressed. I am willing to amend my score, if the questions are properly addressed.

---

## Post-rebuttal updates:

Since the rebuttal has partially addressed my concerns, I am now more inclined to accept the paper and have raised my **Recommendation** score to 6. However, due to some of the issues outlined in my response to the rebuttal, I still find this paper on the borderline. See [here](https://openreview.net/forum?id=b9tUk-f_aG&noteId=8vp4F8U2You) for details.

This paper presents an interesting topic, yet further investigation is needed to answer the lingering questions. The authors should ensure a more comprehensive exploration in the camera-ready version.

---

> ### Author Response · Authors · 2022-11-16
> **Rebuttal response (1/2)**
>
> We thank the reviewer for their constructive comments and suggestions. We will address the concerns in the following.
>
> > Lack of evaluation of the representation quality
>
> In this work, our focus was mainly to provide a large step towards letting object-centric learning methods discover objects on real world data. We note that recent work on object-centric learning, e.g. SLATE, Genesis-v2 [1], SAVi [2], SAVi++ [3], STEVE [4], also do not evaluate the quality of the learned representations. However, we agree that such an evaluation would be interesting. Setting up such an experiment will take time, but we will try to incorporate it by the end of the week (deadline for changes to the paper) or in the camera ready version if it is not ready on time.
>
> > The quantitative results of the baseline model SLATE are questionable. [...] we have also conducted a similar experiment on various synthetic and real-world datasets, and SLATE outperforms slot attention on all datasets for object segmentation and could achieve higher scores than that reported in the paper for the MOVI datasets.
>
> First, note that we report results from a stronger version of Slot Attention that uses a ResNet encoder, which could partially explain the higher performance of Slot Attention. In order to verify our SLATE baseline’s performance, we reran SLATE using the official implementation, and get the following results (1 seed):
>
> - MOVi-C, 11 slots: ARI 42.6, mBO 23.1
> - MOVi-E, 11 slots: ARI 50.0, mBO 17.7
> - MOVi-E, 24 slots: ARI 40.9, mBO 20.1
>
> Results on PASCAL and COCO are still in progress and will be reported once they are ready. Comparing this with the results from our re-implementation:
>
> - MOVi-C, 11 slots: ARI 42.3, mBO 19.5
> - MOVi-E, 11 slots: ARI 47.5, mBO 14.1  [Table 10 in our paper]
> - MOVi-E, 24 slots: ARI 41.9, mBO 17.8
>
> We find that the official implementation gets somewhat better results, potentially due to the more involved scheduling of learning rates based on the validation loss evolution that the official implementation does. Moreover, we discovered that the STEVE paper [4] (also by SLATE’s authors) reports results on image-based MOVi-E, yielding an ARI of roughly 48%, with 15 slots [5].
>
> We also want to highlight that the results on MOVi depend a lot on the number of slots used (which we already reported in Table 10, with now added qualitative results in Figure 17), and that SLATE seems to be sensitive to the number of slots used (with our own method improving even more when reducing the number of slots). This could explain some of the differences to the results the reviewer observed in his own experiments.
>
> This suggests to us that the numbers we reported previously were not uncharitable towards SLATE. However, to represent SLATE in the best possible way, we will replace the numbers with the numbers from the official implementation once we have gathered 5 seeds. If you have any suggestions that could further improve SLATE’s numbers (e.g. code repository, hyperparameters), we would be open to integrating them. Big picture, our interpretation of the results stays the same: SLATE (like Slot Attention) struggles on the MOVi datasets, and our method presents a significant improvement.
>
> > The corresponding qualitative evaluations are also missing. Please include them in the qualitative evaluation.
>
> We now include SLATE example predictions on MOVi in Figures 15, 16, 17 in the appendix (because of space issues in the main part), already from the improved runs using the official implementation. We will also update the paper with COCO examples once they are ready.
>
> > The claim of the method is to achieve object-centric learning in real-world images. However, the method seems to provide only semantic-level segmentation on real-world (or possibly all) datasets.
>
> We want to highlight that our method is capable of doing instance-level grouping on real-world data, at least to some degree. In particular, there is a difference between the MLP and the Transformer decoder, with the MLP decoder actually grouping into instances often (see Appendix Figures 19, 21, 22).
>
> We think the examples we presented in Figure 3 (using the Transformer decoder) do not do the best job in conveying the instance-grouping behavior. To highlight this important detail about our method, we added the new Figure 5, which clearly demonstrates the differences in grouping behavior between MLP and Transformer decoder. Generally, we think this is an interesting aspect of our paper which to our knowledge has not been studied before. While instance-level grouping does not yet work perfectly, this is also to be expected when moving to complex real-world scenes presenting high levels of ambiguities. We think our approach makes a reasonable first step that can be built upon by future work.

---

> > ### Author Response · Authors · 2022-11-16
> > **Rebuttal response (2/2)**
> >
> > > Since the MOVI dataset rarely contains multiple same-class objects in one image, i.e., instance-level segmentation and semantic-level segmentation might have similar results. Therefore, the model could actually be doing semantic-level grouping on all datasets. Please verify this part.
> >
> > We now added an experiment in Appendix B.6 to check whether our method performs semantic grouping on MOVi. In particular, we generated two new variants of the MOVi-C dataset: in the first variant, only objects of the same semantic category are spawned for each scene. In the second variant, the same object is duplicated several times for each scene. We evaluate our model on these variants, finding that performance drops only slightly. This, together with the qualitative evaluation (Table 15), shows that our method does in fact perform instance-level segmentation on MOVi.
> >
> > > Semantic segmentation baselines should also be included in the MOVI experiment
> >
> > Thank you for this suggestion. We now include results of the state-of-the-art semantic segmentation method STEGO [6] on the MOVi datasets in Appendix B.7. We find that STEGO achieves an ARI of around 40% on the MOVi datasets, which is less than the Slot Attention and SLATE baselines (Table 16). Qualitatively, STEGO is able to segment foreground objects and background to some degree, but does not achieve a split between objects (Figure 11). Compared to our method, STEGO’s masks are a lot less consistent on more complex scenes/objects.
> >
> > > Is there a reason why mBO is used instead of mIoU which involves object matching and could potentially provide a more accurate evaluation of the segmentation quality?
> >
> > As we are in the unsupervised setting, computing mIoU will always require some way of matching predictions to ground truth. mBO uses one particular form of matching (for each ground truth mask, match the predicted mask with the highest IoU) and is thus a form of mIoU. In our experience, it is a more accurate judge of mask quality than ARI. Note that while recent video object-centric papers (SAVi 2], SAVi++ [3]) also report mIoU, this is only possible because they assign each slot to a ground truth mask in the first frame.
> >
> > ## References
> >
> > [1]: Engelcke, M., Jones, O.P., & Posner, I. (2021). GENESIS-V2: Inferring Unordered Object Representations without Iterative Refinement. NeurIPS 2021.
> >
> > [2]: Kipf, T., Elsayed, G.F., Mahendran, A., Stone, A., Sabour, S., Heigold, G., Jonschkowski, R., Dosovitskiy, A., & Greff, K. (2022). Conditional Object-Centric Learning from Video. ICLR 2022.
> >
> > [3]: Elsayed, G.F., Mahendran, A., Steenkiste, S.V., Greff, K., Mozer, M.C., & Kipf, T. (2022). SAVi++: Towards End-to-End Object-Centric Learning from Real-World Videos. NeurIPS 2022.
> >
> > [4]: Singh, G., Wu, Y., & Ahn, S. (2022). Simple Unsupervised Object-Centric Learning for Complex and Naturalistic Videos. NeurIPS 2022.
> >
> > [5]: https://github.com/singhgautam/steve/issues/2
> >
> > [6]: Hamilton, M., Zhang, Z., Hariharan, B., Snavely, N., & Freeman, W.T. (2022). Unsupervised Semantic Segmentation by Distilling Feature Correspondences. ICLR 2022.

---

> > > ### Comment · Reviewer_YkFD · 2022-12-05
> > > **RE:Rebuttal**
> > >
> > > I thank the authors for the detailed feedback and the additional experiments provided in the revision. I am increasing the score of the **Recommendation** to 6 as the following concerns have been (partially) addressed:
> > >
> > > 1. The revision provides an evaluation of the representation quality
> > > 2. The revision shows improved baseline results of SLATE by using the official implementation
> > > 3. The revision provides an analysis of semantic vs instance segmentation on a MOVI dataset on both DINOSAUR and one semantic segmentation baseline (STEGO)
> > >
> > > However, I still have the following concerns I hope the camera-ready version of the paper can address (if the paper gets accepted)
> > >
> > > 1. It appears that the baseline models use much simpler encoders in comparison to the proposed model. As the authors show in the revision and rebuttal, changing the architecture of the baseline models improves the performance. This makes me curious if the performance gain of the proposed model is coming from the higher model capacity instead of reconstructing the features. An additional comparison where all models use a comparable size of an encoder is required to answer this question.
> > > 2. Comparing the proposed model to baseline models on the two synthetic MOVI datasets is not a fair comparison, as the proposed model (the DINO encoder) was pre-trained on much larger natural image datasets. To make for a fairer comparison, we should pre-train the DINO model on each MOVI dataset. Please provide further insights or additional experiments in the paper.
> > >     - My initial review did not present the previous two concerns, so the authors could not address them before the deadline. It is thus not necessary for the revision. However, these issues are linked to the core hypothesis and should be addressed at a later stage.
> > > 3. In the revision, only STEGO is evaluated in the MOVI dataset. We suggest that the camera-ready version consider evaluating other baseline models in Table 4, along with STEGO in the MOVI dataset.

---

> > > > ### Author Response · Authors · 2022-12-07
> > > > **Re**
> > > >
> > > > We thank the reviewers for increasing his score of our paper! We would like to respond to his three further points:
> > > >
> > > > 1) We already report numbers for our method with a ResNet34 backbone in Appendix Table 12, which is the exact backbone that we use for the Slot Attention baseline. As can be seen, this does not significantly change the results, showing that the performance gain indeed comes from feature reconstruction. Regarding SLATE, while it's backbone could in principle also be changed to e.g. a ResNet34, we believe it's architecture is closely tied to the VQ-VAE (e.g. SLATE's slot attention module operates on discrete tokens) and thus changing this part would no longer be a faithful representation of the method.
> > > >
> > > > > As the authors show in the revision and rebuttal, changing the architecture of the baseline models improves the performance.
> > > >
> > > > We are slightly confused about this point, as we believe we have not altered any architectural components of our baselines (Slot Attention, SLATE) in the rebuttal.
> > > >
> > > > 2) We agree that the role of the pre-training dataset is an under-discussed aspect in the paper and will try to address this in the camera-ready version. We see including additional data as a strength of our contribution that was so far never considered in previous object-centric learning methods (to our knowledge).
> > > >
> > > > 3) The reviewers initial concerns for including semantic segmentation baselines on MOVi were a) our method does semantic-level grouping and thus should be compared to semantic segmentation methods, b) it would demonstrate "if the existing unsupervised semantic segmentation models already solve the object grouping problem in the multi-object datasets" and c) it would "reveal some limitations of existing synthetic multi-object datasets". For a), we believe we have shown that our method is capable of instance-level grouping and so this point seems to no longer apply. For b) and c), we used the state-of-the-art method STEGO and have shown that it is *not* able to solve the object grouping problem. While it could be interesting to investigate these questions further, we think including more semantic segmentation methods becomes somewhat out-of-scope for the paper. Thus, we would like to kindly ask if the reviewer sees any other reason for evaluating more semantic segmentation baselines, or any further specific insights that this comparison would reveal?

---

> > > > > ### Comment · Reviewer_YkFD · 2022-12-08
> > > > > **Re:Re**
> > > > >
> > > > > 1. Model capacity matching
> > > > >     - I am also slightly confused. I concluded that the architecture of the baselines in the papers is modified from the original implementation because, in the rebuttal, the authors state that both slot attention and SLATE were implemented differently from the official design. And It appears that changing the architectures led to different quantitative results, as indicated by the assumption from the rebuttal that ResNet slot attention has superior performance and also by how reverting back to SLATE's official implementation improved results. It is thus necessary to provide a further investigation into these models’ architectural designs by matching their model capacity to the proposed method.
> > > > >
> > > > >     - And I disagree with the assertion that it is unnecessary to test bigger backbones for SLATE for it is closely tied to the VQ-VAE. My initial question may have been misleading, as the encoder isn't the only component that needs adjustment. The key is to match the model capacity, which can involve increasing both the VQ-VAE and SLATE's decoder size.
> > > > >
> > > > > 1. Including other CV baselines on MOVi
> > > > >     - The reason for proposing the question is that STEGO is not suitable for this task by design, as it performs dataset-level feature clustering. However, other methods like Deep Spectral Clustering, which does image-level feature grouping, might have the potential to solve MOVI dataset. So, it would be interesting to see their performance on MOVI. But since the semantic segmentation concern has been addressed in Appendix B.6 I do agree that this suggestion is somewhat out-of-scope.

---

> > > > > > ### Author Response · Authors · 2022-12-12
> > > > > > **Regarding architectures**
> > > > > >
> > > > > > > I concluded that the architecture of the baselines in the papers is modified from the original implementation
> > > > > >
> > > > > > - For Slot Attention, we replaced the CNN encoder with a ResNet34 encoder (as shown to produce stronger results in SAVi). This was in the initial version of the paper and did not change during the rebuttal. We show that our method out-performs Slot Attention using the exact same ResNet-34 encoder (Appendix Table 12)
> > > > > > - For SLATE, we used the original architecture but re-implemented it in our own codebase. During the rebuttal, we moved to the official implementation (**not** changing the capacity of the architecture), which produced better results for two reasons:
> > > > > >   - It uses different learning rates for the VQ-VAE and the remaining parts of the model.
> > > > > >   - We stop training when the performance of SLATE on the test set starts to degrade. As SLATE's performance drops quite a bit while training goes on, this has a significant impact on the final numbers.
> > > > > >
> > > > > > > The key is to match the model capacity, which can involve increasing both the VQ-VAE and SLATE's decoder size.
> > > > > >
> > > > > > We can indeed do this for the camera-ready version, although we believe it will not significantly change the results:
> > > > > > - The patches the VQ-VAE (independently) processes are only 4x4 pixels large, so scaling this part up by using more layers/features seems unlikely to result in better discrete tokens as the amount of input information is already quite small.
> > > > > > - SLATE's Transformer decoder is already quite heavy with 8 layers. In our experience, equipping the Transformer decoder with too much capacity can easily lead to failed training/not discovering objects anymore.

---

> ### Author Response · Authors · 2022-11-18
> **Evaluation of representation quality and full SLATE results**
>
> Dear reviewer,
>
> we now added an experiment evaluating the quality of the learned slot representations (Appendix B.8, Fig. 12). In particular, we test object property prediction on COCO. We find that our method allows for reasonable downstream performance, performing overall better than the Slot Attention baseline.
>
> We also updated the paper with the SLATE results using the official implementation. We found that the results of SLATE noticeably degraded with longer training and thus picked a lower number of training steps. With these changes, we find that SLATE performs on-par with Slot Attention on the MOVi datasets, and reaches markedly better results than Slot Attention on PASCAL VOC and COCO. On COCO, qualitative results show that semantic grouping starts to emerge for SLATE (Appendix Fig. 15): background like sky, grass and roads are separated out; for foreground objects, more local patterns are sometimes captured correctly. We think there is still a large and clear difference to our method, but we adapted the discussion of the experiments to reflect these new results.
>
> If we have addressed all concerns of the reviewer, we would kindly ask him to adapt his score of the paper.

---

### Author Response · Authors · 2022-11-16
**Major Revision Uploaded**

We thank the reviewers for their time and thorough feedback. We briefly summarize shared points among the reviewers and summarize how we updated the paper to address said concerns. We go into more detail in the individual responses to the reviewers.

Reviewers YkFD, 5iSb and gqpC: Further quantification / explanation of the instance level vs. semantic level grouping trade-off

- We address this by generating a new variant of the MOVi-C dataset, where images contain multiple occurrences of the same object or the same object class. In general, DINOSAUR is also able to perform instance level grouping despite the same object / class being present in the image multiple times. Detailed results can be found in Appendix B.6. We further added a new Figure (Figure 5) in order to elucidate the different grouping behavior of our approach with different decoders in the main part of the paper.

Reviewers 5iSb, fW5l and gqpC: Evaluation of alternative self-supervised pretraining strategies

- To address this point, we ran our approach additionally using features from self-supervised methods MoCo-v3, MSN and MAE and included the results into Table 6 of section 4.4. Interestingly, it shows that the proposed approach is fairly robust to the exact selection of self-supervised pretraining method.

Additionally we added the following results to the paper to address the concerns of individual reviewers:

- A study of training a ResNet encoder from scratch on all datasets (App. B.3, Table 12), also serving as a study of the role of the encoder backbone
- Unsupervised semantic segmentation baseline STEGO on MOVi datasets (App. B.7)
- More example predictions on all datasets (App. E), including from the new self-supervised methods and the SLATE baseline.

We also improved the writing in many places, taking into account the new results and the reviewer’s feedback.

---

### Author Response · Authors · 2022-11-18
**New Revision with Experiment Evaluating Slot Representations**

Dear reviewers,

we posted another revision of the paper, adding an experiment evaluating the learned slot representations for the downstream task of object property prediction (Appendix B.8, Figure 12). We also include improved results for the SLATE baseline. Finally, we again improved the writing in a few places and added some minor experimental details.

---

### Author Response · Authors · 2022-12-07
**Comparison to video methods on MOVi and KITTI results**

Dear reviewers,

after discussion with reviewer fW5L, we trained the video-based methods SAVi++ [1] and STEVE [2] on the MOVi datasets, and evaluated them on a frame-by-frame basis (i.e. image-based). We report foreground image ARI in the following table (single seed).

| Method                       | MOVi-C, 11 Slots | MOVi-E, 11 Slots | MOVi-E, 24 Slots |
|------------------------------|:----------------:|:----------------:|:----------------:|
| SAVi++, Image Reconstruction  | 10.3             | 33.8             | 17.4             |
| SAVi++, Optical Flow         | 29.8             | 16.8             | 19.8             |
| STEVE                        | 49.4             | 51.8             | 47.5             |
| DINOSAUR                     | 68.9             | 79.2             | 65.1             |

*Discussion*: SAVi++ does not perform well without the conditioning signal and temporal context. On MOVi-E, optical flow supervision does not help, most likely because of the camera movements. STEVE performs slightly better than it's image-based counterpart SLATE. There is a considerable gap to our method. More seeds might change the results for SAVi++ somewhat as it exhibits some training variance, but would not change the overall trend.

*Details*: For SAVi++, we use the official code release and train both with image reconstruction and (ground truth) optical flow prediction, but without conditioning signal in the first frame. Besides using a batch size of 32 instead of 64 (because of limited GPU memory), all other hyperparameters are the same as used by SAVi++ on the MOVi datasets. For STEVE, we also use the official code release and the exact hyperparameters as used on the MOVi datasets. We evaluate both models after 200k train steps. The training curves did not indicate that more training would significantly change the results.

### KITTI Dataset

Additionally, we ran our method on the KITTI dataset for autonomous driving in order to compare to the video method from Bao et al. [3]. We report foreground image ARI in the following table (single seed for DINOSAUR, Bao et al's result taken from [3]).

| Method  |   KITTI  |
|------------|:----------:|
| Bao et al. | 47.1 |
| DINOSAUR | 70.1 |

*Discussion*: There is a large gap to Bao et al's method, even though they use supervised methods to estimate optical flow and to extract a motion segmentation from optical flow, and supervise their attention masks using the motion segmentation. In contrast, our method uses no additional supervision and no motion cues.

*Details*: We use a DINO ViT-B/8 backbone and 9 slots. Because KITTI has high resolution images, we chunk the image into 4 crops and rejoin the resulting 4 masks (with then 4*9=36 slots) before computing ARI against the high-resolution ground truth mask. Note that KITTI's instance segmentation annotations are image-based, and so the number from Bao et al. can directly be compared to ours.

### Conclusion

When comparing to video-based object-centric methods *on images*, our method clearly shows better performance. This demonstrates that the reconstruction of strong self-supervised features used in our approach is a valuable signal for the grouping of objects that works in the absence of temporal context. Note that we do not want to make the claim that feature reconstruction is overall 'better' than motion signals for object-centric learning, but rather see the two signals as complimentary. Moreover, the results on KITTI are more evidence that our method works on real-world datasets.

### References

[1]: Elsayed, G.F., Mahendran, A., Steenkiste, S.V., Greff, K., Mozer, M.C., & Kipf, T. (2022). SAVi++: Towards End-to-End Object-Centric Learning from Real-World Videos. NeurIPS 2022.

[2]: Singh, G., Wu, Y., & Ahn, S. (2022). Simple Unsupervised Object-Centric Learning for Complex and Naturalistic Videos. NeurIPS 2022.

[3]: Bao, Z., Tokmakov, P., Jabri, A., Wang, Y., Gaidon, A., & Hebert, M. (2022). Discovering Objects that Can Move. Conference on Computer Vision and Pattern Recognition (CVPR).

---

### Decision · Program_Chairs · 2023-01-20

**Decision:**

Accept: poster

**Justification For Why Not Higher Score:**

1. Still many limitations when evaluating real-world scenes.
2.  The clarity needs to be further improved.

**Justification For Why Not Lower Score:**

Reasonable good results on real-world scenes.

**Metareview: Summary, Strengths And Weaknesses:**

Four experts reviewed this paper with mixed scores (2 accept, 1 borderline reject). AC does feel that this work takes a meaningful step towards real-world object-centric learning by introducing a few new techniques. The reviewers did raise some valuable concerns. The authors are encouraged to make the necessary changes in the camera-ready version.



**Note From Pc:**

if the above contains the word "oral" or "spotlight" please see: "oral" presentation means -> notable-top-5% and "spotlight" means -> notable-top-25%. As stated in our emails, we are disassociating presentation type from AC recommendations

**Summary Of Ac-Reviewer Meeting:**

All reviewers (including the negative reviewer) agree that the paper has some merits and deserve to be presented at ICLR.